# Human and climatic drivers affect spatial fishing patterns in a multiple-use marine protected area: The Galapagos Marine Reserve

Mauricio Castrejón[1¤]*, Anthony Charles[2]

**1** Interdisciplinary PhD program, Dalhousie University, Halifax, Nova Scotia, Canada, **2** Management Science & Environmental Science, Saint Mary's University, Halifax, Nova Scotia, Canada

¤ Current address: Faculty of the Environment, University of Waterloo, Waterloo, Ontario, Canada.
* mauricio.castrejon@dal.ca

**Data Availability Statement:** The Galapagos National Park Service is the owner of the data used for this paper. Therefore, there are legal restrictions

## Abstract

Assessments of the effectiveness of marine protected areas (MPAs) usually assume that fishing patterns change exclusively due to the implementation of an MPA. This assumption increases the risk of erroneous conclusions in assessing marine zoning, and consequently counter-productive management actions. Accordingly, it is important to understand how fishers respond to a combination of the implementation of no-take zones, and various climatic and human drivers of change. Those adaptive responses could influence the interpretation of assessment of no-take zone effectiveness, yet few studies have examined these aspects. Indeed, such analysis is often unfeasible in developing countries, due to the dominance of data-poor fisheries, which precludes full examination of the social-ecological outcomes of MPAs. In the Galapagos Marine Reserve (Ecuador), however, the availability of long-term spatially explicit fishery monitoring data (1997–2011) for the spiny lobster fishery allows such an analysis. Accordingly, we evaluated how the spatiotemporal allocation of fishing effort in this multiple-use MPA was affected by the interaction of diverse climatic and human drivers, before and after implementation of no-take zones. Geographic information system modelling techniques were used in combination with boosted regression models to identify how these drivers influenced fishers' behavior. Our results show that the boom-and-bust exploitation of the sea cucumber fishery and the global financial crisis 2007–09, rather than no-take zone implementation, were the most important drivers affecting the distribution of fishing effort across the archipelago. Both drivers triggered substantial macro-scale changes in fishing effort dynamics, which in turn altered the micro-scale dynamics of fishing patterns. Fishers' adaptive responses were identified, and their management implications analyzed. This leads to recommendations for more effective marine and fishery management in the Galapagos, based on improved assessment of the effectiveness of no-take zones.

on sharing a de-identified data set. However, the data underlying the results presented in the study are available on request at investigacion@galapagos.gob.ec. We confirm that other researchers would be able to access the data set in the same manner as we did, and we did not have any special access privileges that others would not have.

**Funding:** MC is grateful for the financial support provided by the Consejo Nacional de Ciencia y Tecnología (CONACYT-Mexico), and the World Wildlife Fund's Russell E. Train Education for Nature Program. AC acknowledges funding support from the Natural Sciences and Engineering Research Council of Canada (NSERC) and the Social Sciences and Humanities Research Council of Canada (SSHRC), through the Community Conservation Research Network (www. communityconservation.net). The funders had no role in study design, data collection and analysis, decision to publish, or preparation of the manuscript.

**Competing interests:** The authors have declared that no competing interests exist.

# Introduction

There is a growing recognition worldwide that marine protected areas (MPAs), in combination with co-management regimes and the allocation of spatially-exclusive fishing rights to local fishing communities, can be an effective solution for rebuilding depleted marine populations and conserving key biodiversity areas [1–4]. This trend has encouraged an increasing number of governments to adopt this spatially-explicit management tool to promote the recovery of small-scale fisheries and conserve marine biodiversity [5,6]. Unfortunately, typically very limited or no human and economic resources are allocated to monitor the performance of MPAs, notably in developing countries [7]. Consequently, in some regions, such as Latin America and the Caribbean, there are few empirical examples that demonstrate the long-term socio-ecological outcomes generated by the adoption of MPAs [7,8], particularly those designed for multiple use (i.e., those MPAs that allow extractive use in a regulated way, generally under marine zoning schemes, which may include no-take zones). On the other hand, even in those cases in which an assessment of the performance of MPAs can be carried out, research is usually biased to the bio-ecological aspects of the coastal social-ecological systems, leading to poor understanding of human (social, economic, cultural and institutional) dimensions that influence the effectiveness of MPAs to accomplish conservation and fishery management objectives [9–11].

Major areas of analysis about human dimensions of MPAs include how fishers deal with their displacement from traditional fishing grounds as a result of MPA implementation, and the management implications associated with adaptation of their fishing patterns (i.e., variations in selection of fishing grounds, fishing methods, target species, organizational and marketing processes) [12–14]. There are also analyses of how, in many cases, fishing effort tends to aggregate around MPA boundaries, an effect known as "fishing the line", indicating that MPA location is of interest to fishers either because they have traditionally fished around those areas or because a spillover of adults, or larval export, to nearby fished areas has occurred [15,16].

While the need to understand the implications of MPAs for fisheries is clear [17,18], it is important to recognize that other factors affect fishing patterns in addition to MPAs. Recent studies suggest that spatiotemporal allocation of fishing effort is not only influenced by the location of MPAs, but also by factors such as distance of fishing grounds to the nearest port, weather and oceanographic conditions, habitat features, fishing method employed, travel costs, product price and expected revenues [14,19,20]. Nevertheless, to our knowledge, no study has yet examined, in a quantitative way, how fishers respond to those situations in which they have to cope simultaneously with implementation of a multiple-use MPA and with diverse climatic and human drivers of change–usually ignored or neglected in MPA management effectiveness assessments–such as extreme climatic events (e.g., El Niño), the globalization of markets, and the boom-and-bust exploitation of alternative fisheries.

Each driver of change can produce "cascade effects" on the socioeconomic dynamics of fishing communities, whether through changes in the availability and accessibility of target species or variations in environmental and market conditions. This leads fishers to adapt their fishing patterns to prevent or mitigate the damage to their livelihoods [12]. If the main reasons behind these adaptations are not well understood, a bias in the interpretation of the observed patterns could be produced, leading to errors in planning, implementing and assessing MPAs. This is relevant for fisheries management, particularly in those cases in which the main assumption in assessing the effectiveness of an MPA is that any adaptation in fishing patterns was caused exclusively by the MPA, rather than by the combined impact of different human and climatic drivers of change.

The Galapagos Marine Reserve (GMR) represents a unique biodiversity and climate change hotspot in Latin America and the Caribbean, which provides an excellent case study to illustrate how the interaction of various large-scale human and climatic drivers around a network of no-take zones influences fishing patterns. In this multiple-use MPA, marine zoning was implemented between 2000 and 2006, in combination with a co-management regime and the allocation of exclusive fishing rights to local small-scale fishers, to mitigate the impacts of human activities on sensitive ecological areas and to ensure the sustainability of Galapagos small-scale fisheries [21,22]. However, decisions to locate no-take zones in areas of low abundance of the most lucrative fishery resources, in combination with a lack of effective enforcement and a high rate of non-compliance [23], severely limited the effectiveness of Galapagos marine zoning for shellfish fisheries management purposes [2]. Despite these shortcomings, spiny lobster (*Panulirus penicillatus* and *P. gracilis*) stocks showed an unexpected and remarkable recovery after a period of overexploitation [24]. Previous studies suggest that the Galapagos spiny lobster fishery recovery was caused by the combined effect of market forces and favorable environmental conditions, rather than no-take zone implementation [2,25]. However, this hypothesis has not been tested yet by a long-term impact assessment of the effectiveness of no-take zones. This type of assessment could be influenced by fishers' adaptive responses, so a proper assessment can only be made given a better understanding of how local fishing communities coped with the interactions of human and climatic drivers, before and after marine zoning implementation.

Using geographic information system (GIS) modelling techniques, in combination with boosted regression models, this paper evaluates how the spatiotemporal allocation of fishing effort in the Galapagos spiny lobster fishery was affected by the interactions of human and climatic drivers over a 15-year period (1997–2011). Based on the analysis of changes in fishing patterns, we build an understanding of the main drivers and factors influencing fishers' adaptive responses to drivers of change, before and after implementation of marine zoning, including their links to the geographic and socioeconomic features of fishing communities, and their implications for fisheries management. We integrated this knowledge to provide a series of recommendations to improve the design and effectiveness of Galapagos marine zoning to reconcile conservation and fishery management objectives.

## Materials and methods

### Study area

The Galapagos Islands is comprised of approximately 234 islands, islets and rocks with a total land area and coastline of ca. 7 985 km$^2$ and 1667 km, respectively [26]. According to Edgar et al. [27], this volcanic archipelago is divided into five marine biogeographical regions, named as far-Northern, Northern, South-Eastern, Western and Elizabeth (Fig 1). Each one shows particular assemblages of fish and macro-invertebrate species, whose abundance and distribution are strongly affected by the El Niño Southern Oscillation [25,28].

Only 4% of the total land area is inhabited by ca. 25,144 residents (Table 1) distributed on five islands (Santa Cruz, Baltra, San Cristobal, Isabela, and Floreana). The remaining land area is protected as a national park. There are three main fishing ports (Baquerizo Moreno, Puerto Ayora and Villamil; Fig 1) that display specific geographic and socioeconomic features, particularly in terms of population density, number of fishers, composition of the fishing fleet, and available land-based tourism infrastructure (Table 1). There are 1084 license holders and 416 vessels registered in Galapagos, although only 37% of them remain active in the spiny lobster fishery (Table 1). Each fishing license provides its owner the right to fish any type of shellfish and finfish species commercially permitted. Approximately 97% of active vessels are smaller

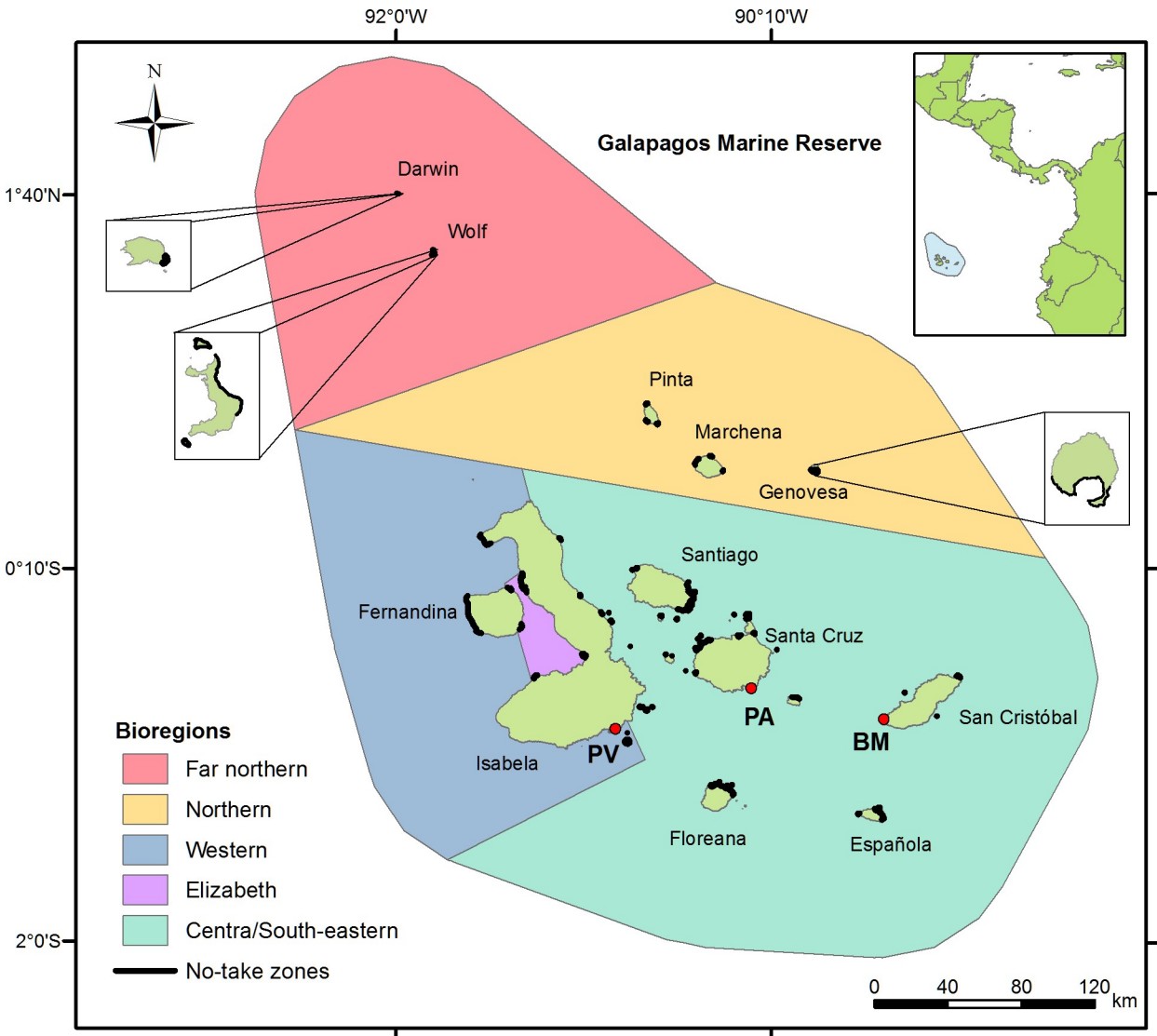

**Fig 1. Marine biogeographical regions of the Galapagos Islands.** Red circles indicate the location of the three main fishing ports: Puerto Villamil (PV), Puerto Ayora (PA) and Baquerizo Moreno (BM). Black areas indicate the location of no-take zones.

than 9.6 m long (fiber glass or wooden made) and equipped with outboard engines (15–200 HP). Only 13% consist of large wooden boats (8 to 18 m long) equipped with inboards engines (30–210 HP). These "mother boats" are used as storage, resting and towing units for up to four small vessels [29]. Most harvesting activities usually last one or two days, although mother boats are able to operate for a maximum of 12 days.

The most valuable shellfish species in Galapagos are the red and green spiny lobsters (*P. penicillatus* and *P. gracilis*), and the sea cucumber *Isostichopus fuscus*, harvested exclusively by artisanal hookah and skin divers mostly in sub-tidal rocky habitats. The fishing season since 1999 for sea cucumbers usually lasts from June to August (two months) and for spiny lobsters from September to December (four months), although slight variations have occurred through the years. The number of landing sites along the coast is quite limited (Table 1), facilitating the systematic and reliable collection of fishery-related data at each port since 1997.

**Table 1. General features of the three main fishing ports of the Galapagos Islands, including a summary of the fishery information analyzed in this study, for each sampling method used.**

| | San Cristobal | Santa Cruz | Isabela | Total |
|---|---|---|---|---|
| Fishing port | Baquerizo Moreno | Ayora | Villamil | 3 |
| Main landing sites | 1 | 2 | 1 | 4 |
| Population[1] | 7495 | 15393 | 2256 | 25144 |
| Coastline (km)[2] | ~ 156 | ~ 170 | ~ 617 | ~ 944 |
| Hotel capacity (beds)[3] | 449 | 990 | 193 | 1632 |
| Restaurants and bars[3] | 35 | 61 | 18 | 114 |
| License holders (active/registered)[4] | 174/552 | 136/293 | 100/239 | 410/1084 |
| Small vessels (active/registered)[4] | 59/163 | 44/87 | 48/107 | 151/357 |
| Mother boats (active/registered)[4] | 2/32 | 1/19 | 1/8 | 4/59 |
| Cooperatives | 2 | 1 | 1 | 4 |
| Interview based data[5] (1997–2011) | 4387 | 4246 | 6727 | 15360 |
| Fishery observer based data[5] (2000–2006) | 1058 | 719 | 586 | 2363 |

[1] Galapagos census 2010 conducted by INEC.

[2] PNG [30].

[3] Epler [31].

[4] Reyes and Ramírez [32].

[5] Participatory Programme of Fisheries Monitoring and Research; no data were collected in 2007.

In March 1998, the Galapagos archipelago and its surrounding open ocean were enclosed in a multiple-use MPA of nearly 138,000 km$^2$, the GMR, through the enacting of the Galapagos Special Law [22,33]. This law decreed an institutional shift from a hierarchical (top-down) to a co-governance mode, and from an open access to a common property regime. Since then, large-scale fishing was prohibited inside the reserve and fishing licenses and permits were exclusively allocated to local small-scale fishers.

The Galapagos' marine zoning was created and implemented between 2000 and 2006 [22]. It comprised 76 no-take zones distributed across the archipelago, covering 17% of the coastline (Fig 1). The dimensions of the zones range from small offshore islets to 22.8 km of coastline [34]. The total area per management zone is unknown, as the offshore boundaries were not legally established [22]. Fishing and tourist activities, such as snorkelling and scuba diving, are prohibited inside 14 no-take zones, known as "conservation zones". In the remaining 62 no-

**Table 2. Main climatic and human drivers that potentially affected the spiny lobster fishery from the Galapagos Islands between 1997 and 2011.**

| Category | Drivers of change | Temporal scale |
|---|---|---|
| Climate and environment | El Niño 1997/1998 | April 1997-June 1998 (~14 months) |
| Governance | Co-governance and common property period | March 1998-onwards |
| International trade and globalization of markets | Boom and bust exploitation of the sea cucumber fishery by roving bandits | April 1999- (decades) |
| Governance | Marine zoning | April 2000-onwards (decades) |
| International trade and globalization of markets | Global financial crisis | December 2007- June 2009 (~18 months) |

take zones, known as "tourism zones", only tourist activities are permitted. No buffer zones were established. Therefore, in some regions, conservation and tourism zones are contiguous, constituting "no-take networks" (i.e., interconnected groups of individual no-take zones). The largest ones are distributed in Fernandina, Santiago, Santa Cruz and Floreana islands (Fig 1).

Before and after creation of the marine zoning, the spiny lobster fishery was impacted by diverse climatic and anthropogenic drivers of change, most of which acted simultaneously on various spatio-temporal scales. The most relevant are indicated in Table 2, which shows the category of the driver, according to the classification defined by Hall [35], the specific form of the change, and the corresponding time scale.

## Fishing effort data

Fishing effort data for the period 1997–2011 were gathered from the Participatory Programme of Fisheries Monitoring and Research (PIMPP, in Spanish). Before this period, there are no spatially-explicit fishing effort data available for analysis. Only annual aggregated fishing effort data (in fishing days) per island for the period 1974–1979, published by Reck [36], remain for comparative purposes.

Fishery-related data were collected from 17,764 fishing trips, equivalent to 20,203 fishing effort records per fishing ground, either by interviewers (1997–2011) or observers onboard (2000–2006), at the three main ports of Galapagos (Puerto Ayora, Baquerizo Moreno, and Villamil) on a daily basis over each fishing season (Table 1 and S1 Table). Geographical positioning systems were usually used by observers to collect spatially-explicit data onboard fishing vessels, including position of fishing grounds, fishing method, effective fishing hours, number of divers, vessel type and name, departure and landing port, departure and arrival date, and catch per spiny lobster species.

The same types of data were collected by interviewers using semi-structured questionnaires. However, in this case, fishing grounds' names visited per fishing trip were obtained instead of the exact geographical location where fishing activity took place. To make this subset of data spatially explicit, we added the geographic coordinates published by Chasiluisa and Banks [37], who defined reference positions (latitude and longitude) for 320 fishing grounds identified and distributed across the archipelago.

The PIMPP dataset was extensively reviewed, standardized and cleaned before being filtered. Data collected by interviewers and observers onboard were included in this study. However, analyses were restricted to those spiny lobster fishing trips conducted in small vessels (fiber glass and wooden made) by one or two hooka divers, where the number of effective fishing days, at a single fishing ground, ranged from one to seven. The final dataset accounts for 17,723 fishing effort data units (Table 1), representing 88% of the original dataset. Approximately 78% of these data are georeferenced, i.e., they include the exact, or reference, position of each fishing ground visited per fishing trip. The remaining 22% of the data simply specify the islands in which fishing activity took place.

To determine the representativeness of the data selected, for each sampling method used, we estimated the sampling effort of small fishing vessels and fishing effort, measured in diver-hours, registered by the PIMPP between 1997 and 2011 (S1 Table). According to our results, sampling effort by interviews was on average 67% and 37% for small fishing vessels and fishing effort, respectively. In contrast, sampling effort by observers on board was 19% and 5% for small fishing vessels and fishing effort, respectively (S1 Table). These results suggest that interview-based data are more representative of the spatiotemporal dynamics of the fishing fleet than data collected by observers onboard. However, they could also be less reliable, if fishers provided inaccurate information about the locations of their fishing grounds. To account for

**Table 3. Periods defined to evaluate the spatio-temporal dynamic of fishing patterns in the spiny lobster fishery, based on the most relevant climatic and human drivers occurring between 1997 and 2011.**

| Period | Acronym | Temporal scale |
|---|---|---|
| 1. Co-governance and El Niño | CoM-EN | June 1997-December 1998 |
| 2. Sea cucumber re-opening phase | RovBan1 | September 1999- December 2000 |
| 3. Sea cucumber expansion phase | RovBan2 | September 2001- December-2002 |
| 4. Sea cucumber overexploitation phase and marine zoning | MarZon | September 2003-December 2005 |
| 5. Sea cucumber collapse phase and global financial crisis | Crisis | September-December 2006 and September-December 2008 |
| 6. Spiny lobster recovery | Recovery | September-December 2011 |

this type of uncertainty, both data sources were analyzed in most cases separately to compare the fishing patterns identified.

## Data analysis

Fishing effort data were grouped into a suitable number of time periods to evaluate how the spatio-temporal dynamics of fishing patterns in the spiny lobster fishery were affected by the potential drivers of change described in Table 2. However, as the temporal scale of each driver is different, and most of them occurred simultaneously, it was not feasible to divide the data available evenly between periods. Accordingly, we defined six time periods (Table 3), based on the following logic:

- We subdivided the boom-and-bust exploitation period of the sea cucumber fishery into four phases (re-opening, expansion, overexploitation and collapse) to evaluate their specific impact on the spatial allocation of fishing effort in the spiny lobster fishery. The sea cucumber overexploitation phase and the marine zoning implementation were grouped together because both occurred simultaneously. In this case, we assumed that marine zoning was implemented after a transition period of three years (2000–2003) once the moratorium on the entry of new fishers was put in place, fishing regulations were decreed, enforcement capacity increased, and fishers were aware of zoning boundaries and legal framework.

- We grouped the sea cucumber collapse phase and the global financial crisis 2007–09 together because both drivers affected the profitability of fishing activity [2,38], allowing us to evaluate their combined impact on fisher's behaviour. Data from 2009 and 2010 were excluded from this period due to a lack of georeferenced data.

- The unexpected and remarkable recovery of the spiny lobster fishery was considered an additional period. Such an event does not represent a driver itself, at least not in the short term, but a social-ecological impact potentially caused by climatic and human drivers that occurred in earlier periods.

**Interaction between sea cucumber and spiny lobster fisheries.** We performed a Pearson's correlation analysis to evaluate how the fishing effort capacity in the spiny lobster fishery was affected by the different phases (re-opening, expansion, overexploitation and collapse) of the boom-and-bust exploitation of the sea cucumber fishery, using as variables the number of active fishers, small-vessels and mother boats in both fisheries between 1997 and 2011. The information was obtained from PIMPP, Galapagos National Park Service fishing registry, Moreno et al. [39] and Reyes and Ramírez [32]. The correlation analysis was conducted in the R statistical programming language, version 3.1.2 (R Development Core Team 2014).

**Spatiotemporal analysis of fishing patterns.** The spatiotemporal allocation of fishing effort across the archipelago was evaluated using GIS modelling techniques with ArcGIS 10.2.2 (ESRI) software. We calculated standard deviation ellipses (SDE) polygons by point pattern statistics [40] to determine the core areas and distribution ranges of the fishing fleets based in the three ports (Baquerizo Moreno, Puerto Ayora and Villamil) during the six periods defined.

In this study, SDE represent graphical summaries of the central tendency, dispersion and directional trends of fishing fleets. Core areas and distribution ranges refer to those areas covering 68% (1 SDE) and 95% (2 SDE) of the full spatial extent of fishing fleet distribution, respectively. Furthermore, to determine if the same areas have been reused by fishers from different ports at different periods, we estimate an index of reuse (IOR), following the procedure described by Morrisey and Gruber [41] and Horta e Costa et al. [42]. Small vessels core areas and distribution ranges were used to estimate IOR, by the following formula [41]:

$$IOR = \frac{OV(A_1 + A_2)}{A_1 + A_2}$$

where [OV ($A_1$+$A_2$)] refers to the overlapping area between two core areas (or distribution ranges), and ($A_1$+$A_2$) to the total area of both core areas (or distribution ranges). IOR values range from 0 (both areas do not overlap) to 1 (both areas overlap completely). One and two-way ANOVAs were employed to test the null hypothesis of absence of differences in core areas, distribution range and IOR between different periods, and between ports and sampling methods (interviews vs fishery observers). A Bartlett's test was performed prior to all analyses to test the assumption of homogeneity of variances among treatments. When data were heteroscedastic, or did not fulfill the normality assumption, transformations were conducted.

We also performed a hotspot analysis using area pattern statistics [40] to evaluate if the areas where most fishing effort is concentrated (i.e., hotspots) have varied across each period and to determine if the fishing patterns identified vary according to the sampling method employed. Based on this analysis, we determined the spatial distribution of hotspots before and after marine zoning implementation, allowing us to evaluate if fishers were displaced from their traditional fishing grounds and if fishing effort concentrates around no-take zones, producing a "fishing the line" effect.

We aggregated fishing effort data per period and sampling method and performed a single hotspot analysis for each possible combination (nine in total). The following procedure was applied to each combination:

1. A grid with a 2.25 km$^2$ cell size was superimposed over the entire archipelago. Such resolution was selected considering the size of the study area, as well as the precision and resolution required to evaluate the fine-scale distribution of fishing fleets;

2. a buffer of 2.5 km was delimited around the coastline of each island, islet and rock, defined based on the dispersion of data and a maximum bathymetry of 40 m, so as to contain the area where the spiny lobster fishery takes place. Grid cells located outside this buffer, including the land area, were removed; then, we proceeded to eliminate those resulting grid cells smaller than 3% of the original grid cell size;

3. total fishing effort (diver-hours) per grid cell was summarized and a measure of effort density (diver-hours km$^{-2}$) was calculated by dividing the total sum of fishing effort per cell by the original grid cell size (2.25 km$^2$);

4. a spatial weights matrix was generated using the k-nearest neighbors (k = 8) as the conceptualization of the spatial relationship among data (i.e., small-vessels). The latter method was

selected considering the extensive and uneven spatial distribution of our data across the study area and the skewed distribution of fishing effort values;

5. finally, a hotspot analysis was performed, using effort density as the input field. Such analysis identified statistically significant spatial clusters of high effort density values (hot spots) and low effort density values (cold spots) across the archipelago, based on the Getis/Ord Gi* statistic [43], producing a Z-score and p-values as measures of statistical significance.

The null hypothesis in this case is that the spatial allocation of fishing effort is the result of random spatial processes, which is rejected if the Z-score ≥ 1.96 and p < 0.05 with a 95% confidence level. A high Z-score and small p-value indicates a hotspot, while a low negative z-score and small p-value indicates a cold spot. The higher (or lower) the Z-score, the more intense the clustering, while a Z-score near zero indicates no apparent spatial clustering [44].

**Climatic and human drivers affecting spatial fishing effort allocation.**    We defined, for the set of fishing georeferenced data, a diverse suite of explanatory variables potentially having an influence on the spatial allocation of fishing effort, measured in diver-hours. Geographic, oceanographic and socioeconomic variables were selected based on the human and climatic drivers identified as relevant for this study. Each was categorized as either temporally static or temporally dynamic, based on whether it changes over time [19].

The first category includes latitude, longitude, bioregion, homeport, distance to home port (DistHP), and distance to the nearest no-take zone (NearNTZ). Here, we defined homeport as the port from which a vessel primarily operates, regardless of its registry. To calculate the shortest effective distance between each fishing record and the corresponding vessel's home port, we conducted a cost-distance analysis using the spatial analyst extension in ArcGIS 10.2.2. The same analysis was used to calculate the shortest effective distance between each fishing record and the nearest no-take zone.

The second category includes historic period (Period), month, average ex-vessel price per year (ExVesPrice), lobster catch obtained in previous fishing trips (PrevCatch), average sea cucumber revenues obtained the fishing season before the beginning of lobster season (SeaCucRev), vessel type, and the Oceanic Niño Index (ONI). The ONI represents the month moving average of ERSST.v3b SST anomalies in the Niño 3.4 region (i.e., west of the GMR 5˚N– 5˚S, 120˚-170˚W), based on centered 30-year base periods updated every 5 years. ONI is the main indicator used by NOAA for monitoring El Niño and La Niña, which are opposite phases of the climate pattern called the El Niño-Southern Oscillation. Data were obtained from the NOAA Climate Prediction Centre at www.cpc.ncep.noaa.gov/products/analysis_monitoring/ ensostuff/detrend.nino34.ascii.txt

Analysis of fishing effort hotspots and fishing fleet distribution ranges and core areas were used to evaluate how spatiotemporal distribution changes in relation to the external drivers analyzed in this study. Then, boosted regression trees (BRTs) were used to identify the factors that explain such spatiotemporal patterns. The goal was to predict fishing effort, measured in diver-hours per port and period, as a function of geographic, oceanographic and socioeconomic variables described above.

BRT models can be defined as flexible additive regression models in which individual terms are simple trees, created by recursive binary splits constructed from predictor variables and combined to optimize predictive performance, which are fitted in a forward, stagewise fashion [45,46]. Unlike general linear models and general additive models, BRT models accommodate missing values in continuous or categorical predictors, are able to handle outliers, collinear variables, interactions between variables, and nonlinear relationships between predictor and response variables, showing additionally similar, or even stronger, predictive performance [19,46,47].

BRT model fitting requires the definition of three parameters: (1) *learning rate* (lr), also known as the shrinking parameter, which determines the contribution of each tree to the growing model (i.e., controlling the rate at which the model converges on a solution); (2) *tree complexity* (tc), which refers to the number of nodes (or splits) in a tree (i.e., the ability of model interactions); and (3) the two previous parameters are used to estimate the *optimal number of trees* (nt) required to increase performance prediction. In addition, to improve accuracy and reduce overfitting, we introduced stochasticity to the BRT model through a "bag fraction", which specifies the proportion of data to be selected at each step [46]. The BRT model was fit to allow interactions using a tree complexity of 2 and a learning rate of 0.005 and a bag fraction of 0.6. Ten-fold, cross-validation of training data was used to determine the optimal number of trees necessary to minimize deviance and maximize predictive performance to independent test data. Model performance was assessed based on predictions made using the independent testing set that was withheld during cross-validation.

Deviance explained and Pearson's correlation coefficient (r) were used to assess the predictive performance of BRT models. Furthermore, variable importance (VI) was estimated by averaging the number of times a variable is selected for splitting and the squared improvement resulting from these splits [48,49]. VI scores provide a measure of the relative influence of predictor variables used to build the model [19]. Values are scaled so that the sum adds to 100, with higher numbers indicating a stronger influence on the response variable. Following Soykan et al. [19], a random number (RN) between 1 and 100 was added to identify useful variables for modeling a response. Useful variables in predicting fishing effort were those that had higher VI scores than RN. Finally, for interpreting BRT models results, we generated a partial dependence plot for each predictor variable. Such graphs show the effect of a variable on the response after accounting for the average effects of all other variables in the model, including the RN [19]. BRT model fitting was conducted in the R statistical programming language, version 3.1.2 (R Development Core Team 2014) using the "gbm" and "dismo" libraries complemented with the brt.functions code developed by Elith et al. [46].

## Results

### Interaction between sea cucumber and spiny lobster fisheries

The analysis of active fishing capacity from 1999 to 2011 showed that large-scale changes in fishing effort dynamics for the spiny lobster fishery occurred during the boom-and-bust exploitation of the sea cucumber fishery and the global financial crisis 2007–09. The active number of fishers, small-vessels and mother boats increased in the spiny lobster fishery, reaching a maximum value during the re-opening and expansion phase of the sea cucumber fishery, corresponding to the RovBan1 and RovBan2 periods (Fig 2A–2C). A similar pattern was observed in the sea cucumber fishery, although in this case the number of fishers and small-vessels reached a maximum during the overexploitation phase of the sea cucumber fishery and marine zoning implementation, corresponding to the MarZon period (Fig 2A–2C). During this period the active fishing capacity in both fisheries showed a strong positive linear trend in Puerto Ayora, Baquerizo Moreno and Puerto Villamil (Fig 2D–2F).

The active fishing capacity in the spiny lobster fishery decreased gradually since 2000 (RovBan1). This trend intensified after the total closure of the sea cucumber fishery occurred in 2006 and during the global financial crisis 2007–09 (Fig 2A–2C). Between 2000 and 2010, the number of fishers decreased 80.3%, from 1183 to 233, while the number of small vessels and mother boats decreased 55.3%, from 286 to 128 (Fig 2A and 2B). However, the most remarkable decrease was observed in the number of mother boats, which decreased 88.0%, from 42 to 5 during the same period (Fig 2C). A similar decreasing trend in active fishing capacity was

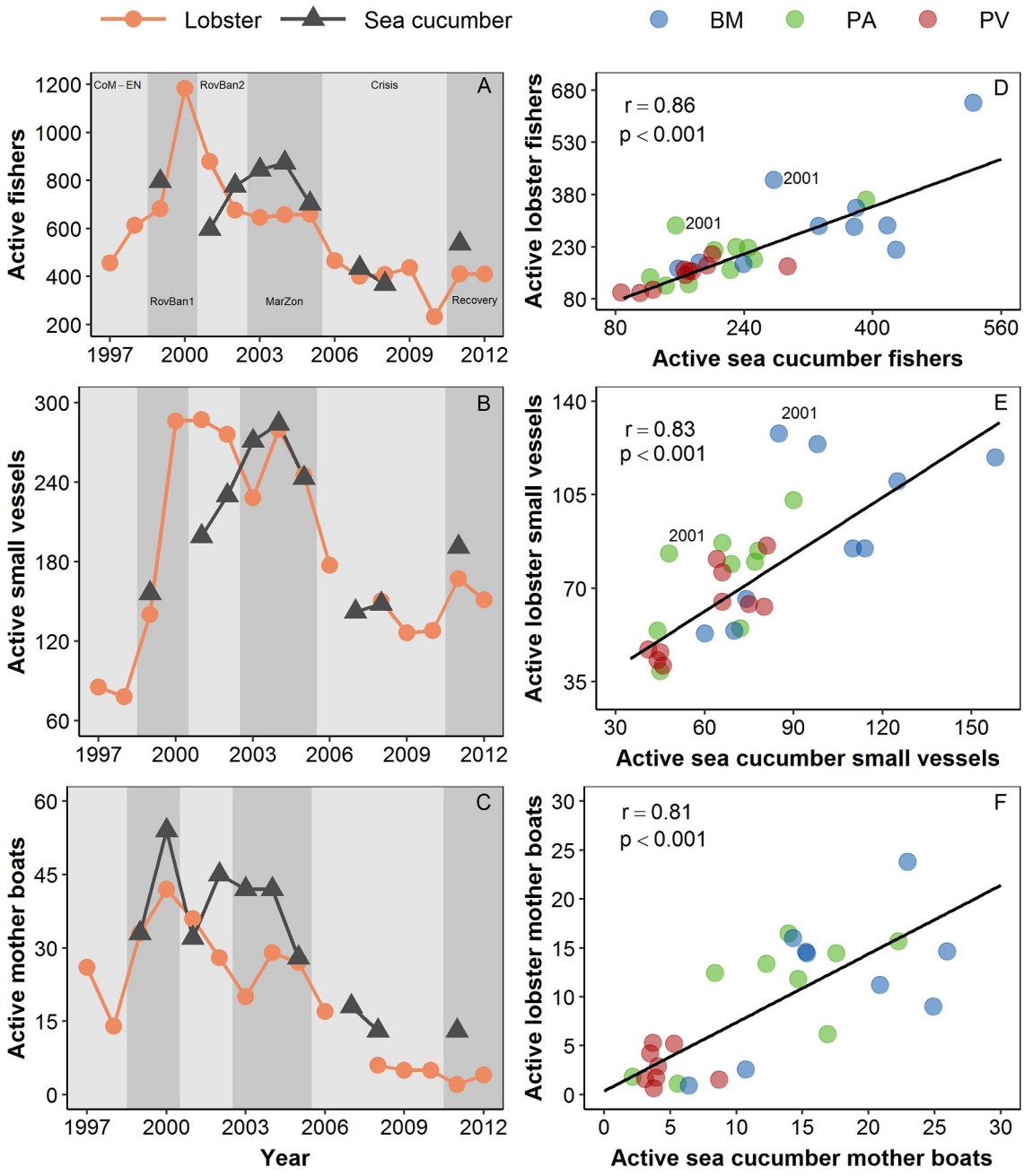

**Fig 2. Long-term variation in fishing capacity in the spiny lobster and sea cucumber fishery from the Galapagos Marine Reserve.**
a) active fishers per year; b) active small-scale vessels per year; c) active mother boats per year; d) relationship between active lobster and sea cucumber fishers per port; e) relationship between active lobster and sea cucumber small-vessels per port; and f) relationship between active lobster and sea cucumber mother boats per port. BM: Baquerizo Moreno; PA: Puerto Ayora; PV: Puerto Villamil; CoM-EN: Co-governance and El Niño; RovBan1: Sea cucumber re-opening phase; RovBan2: Sea cucumber expansion phase; MarZon: Sea cucumber overexploitation phase and marine zoning; Crisis: Sea cucumber collapse phase and global financial crisis; Recovery: Spiny lobster recovery.The sea cucumber fishery was closed five years. In 2001, there was an unsuccessful attempt to implement an individual quota system in the sea cucumber fishery, which led to a temporal reduction in fishing effort [50,51].

observed in the sea cucumber fishery, which was reopened in three occasions after 2006 (Fig 2A–2C). The total number of fishers and small vessels in both fisheries increased slightly during the recovery period of the spiny lobster fishery (Fig 2A and 2B), while the number of

mother boats remained at very low numbers (Fig 2C). These results suggest that a significant number of fishers from Puerto Ayora, Baquerizo Moreno and Puerto Villamil responded to the economic perturbations caused by the collapse of the sea cucumber fishery and the global financial crisis 2007–09, by abandoning the spiny lobster and the sea cucumber fisheries. This macro-scale change in fishing capacity influenced the micro-scale spatiotemporal dynamic of fishing patterns in the spiny lobster fishery, as described in the next section.

### Spatiotemporal analysis of fishing patterns

The spatiotemporal allocation of fishing effort showed different patterns between ports and periods. These did not vary, in most analyses, according to the sampling method used (Figs 3 and 4). According to port-based interviews, Puerto Ayora and Baquerizo Moreno's fishing fleets showed larger core areas and distribution ranges than Puerto Villamil (S2 Table; Fig 3). However, such differences were not significant between ports (core areas: H = 2.667; d.f. = 2; p = 0.264; distribution ranges: H = 1.906; d.f. = 2; p = 0.385). Similar results were shown by observer onboard data (core areas: H = 5.955; d.f. = 2; p = 0.051; distribution ranges: H = 5.067; d.f. = 2; p = 0.079, although in this case Baquerizo Moreno's fishing fleet showed larger core areas and distribution ranges than Puerto Ayora (S2 Table; Fig 4).

Both sampling methods showed that Galapagos fishing fleets had, on average, low degrees of overlap in their fishing activity spaces, particularly in relation to their core areas, as it was denoted by IOR values close to zero, meaning that fishing spaces between fishing fleets do not overlap (S3 Table). Puerto Ayora and Baquerizo Moreno's fishing fleets had the highest similarity of core areas and distribution ranges in comparison with any other combination of ports (S3 Table, Figs 3 and 4). In contrast, Baquerizo Moreno and Puerto Villamil's fishing fleets showed the lowest degree of overlapping of their core areas and distribution ranges (S3 Table, Figs 3 and 4). Nevertheless, both sampling methods showed that the central tendency, dispersion and directional trends of the Baquerizo Moreno, Puerto Ayora and Puerto Villamil fishing fleets' core areas and distribution ranges showed large variations among periods, changing from no overlap to large overlap (Figs 3 and 4).

According to port-based interviews, Baquerizo Moreno fishing fleet's core area and distribution range were located exclusively around San Cristobal Island from 1997 to 2005, corresponding to CoM-EN, RovBan1, RovBan2 and MarZon periods (Fig 3). However, the core area expanded towards Santa Fe, western and southern parts of Santa Cruz and Santiago Islands during the Crisis and Recovery periods, overlapping with Puerto Ayora's fishing fleet core area and distribution range (Fig 3). Baquerizo Moreno's fishing fleet distribution range showed a similar but larger expansion pattern toward Española and the eastern part of Isabela Island, reaching the western part of Floreana during the recovery of the spiny lobster fishery (Fig 3). Observer onboard data showed a similar pattern, although in this case Baquerizo Moreno fishing fleet's core area and distribution range showed a larger expansion during the RovBan2 period (Fig 4). However, both sampling methods showed that spiny lobster fishing grounds located along San Cristobal Island are used exclusively by fishers from Baquerizo Moreno, although Puerto Ayora's fishing fleet did expand temporally part of its distribution range toward San Cristobal during RovBan1 and RovBan2 periods (Fig 3).

Puerto Ayora fishing fleet registered also a large variation in its spatiotemporal distribution between 1997 and 2011 (Figs 3 and 4). According to port-based interviews, the core area and distribution range of this fishing fleet covered exclusively Santa Cruz and Santa Fe Islands during the CoM-EN period, showing no overlapping positions with Puerto Villamil and Baquerizo Moreno (Fig 3). However, Puerto Ayora's core area expanded to Santiago and the west and eastern parts of Isabela Island, while the distribution range extended practically to the entire

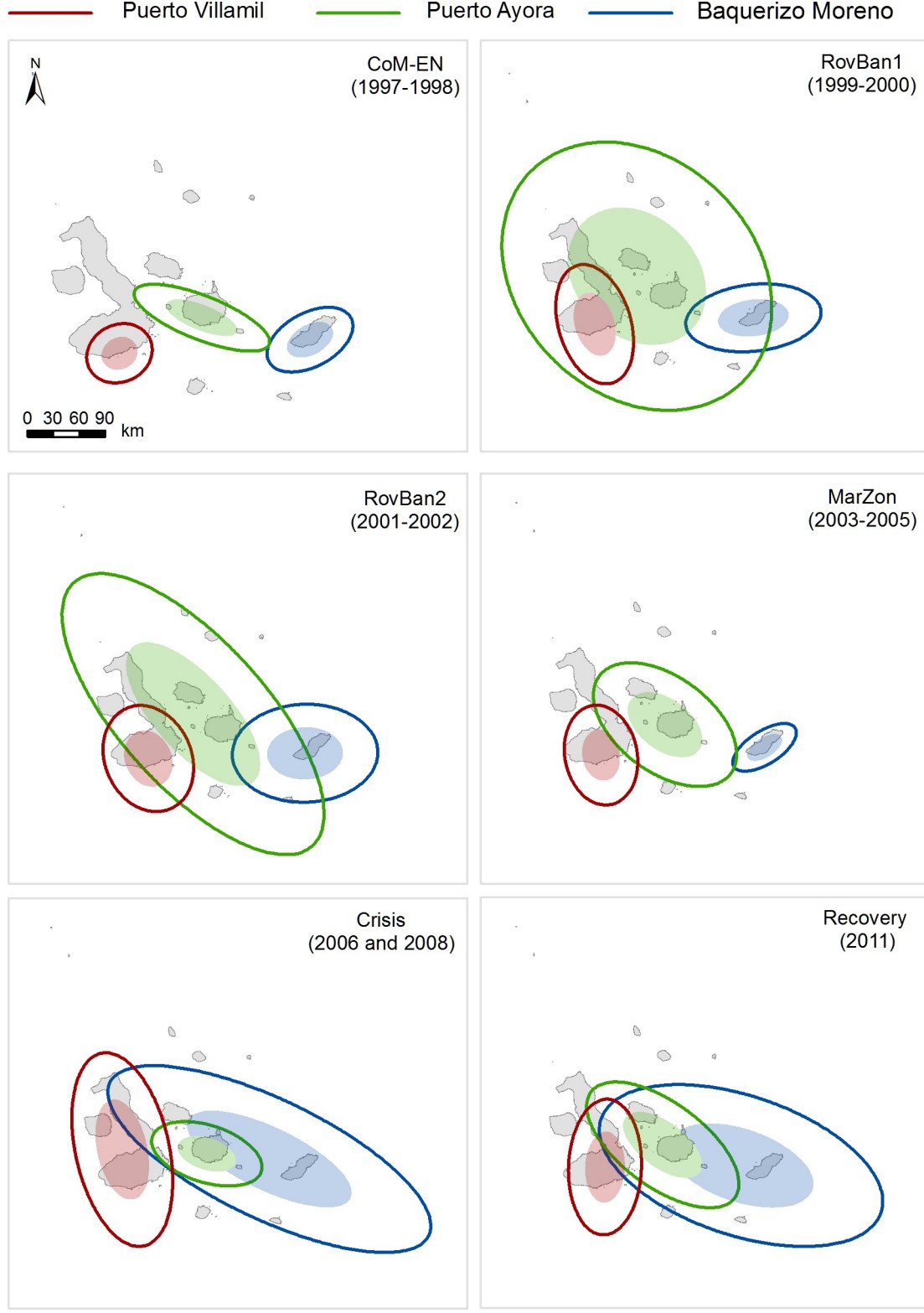

**Fig 3. Core areas (filled ellipses) and distribution ranges (unfilled ellipses) of the fishing fleets from Puerto Ayora, Baquerizo Moreno and Puerto Villamil in the Galapagos Marine Reserve between 1997 and 2011, based on port interview data, for each of the six time periods.** Co-governance and El Niño (CoM-EN); Sea cucumber re-opening phase (RovBan1); Sea cucumber expansion phase (RovBan2); Sea cucumber overexploitation phase and marine zoning (MarZon); Sea cucumber collapse phase and global financial crisis (Crisis); and Spiny lobster recovery (Recovery).

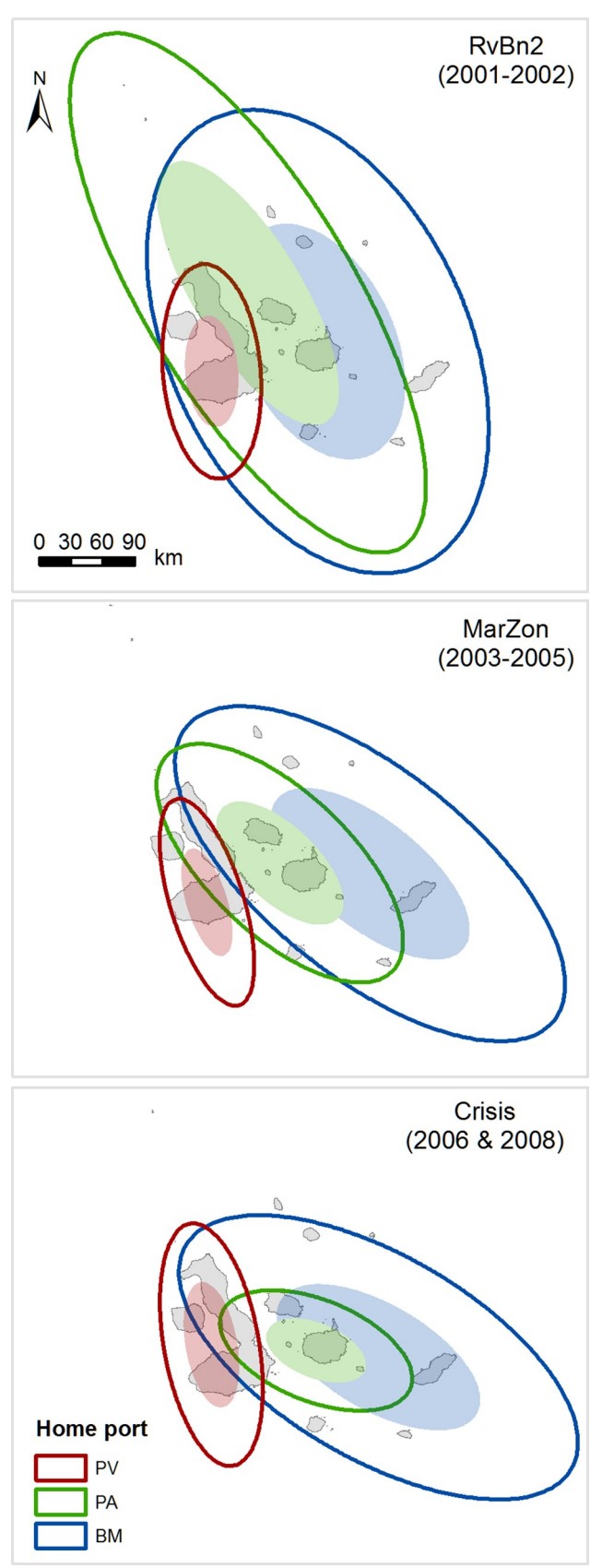

**Fig 4. Core areas (filled ellipses) and distribution ranges (unfilled ellipses) of the fishing fleets from Puerto Ayora, Baquerizo Moreno and Puerto Villamil in the Galapagos Marine Reserve between 2001 and 2008, based on observer onboard data.** Results are shown for three time periods: Sea cucumber expansion phase (RovBan2); Sea cucumber overexploitation phase and marine zoning (MarZon); Sea cucumber collapse phase and global financial crisis (Crisis).

archipelago during the RovBan1 and RovBan2 periods (Fig 3). A similar pattern was shown by observer onboard data, although in this case the core area extended until the far-northern islands of Darwin and Wolf during the RovBan2 period (Fig 4). In contrast, the core area and distribution range contracted remarkably during MarZon and Crisis periods, until reaching similar dimensions to those observed during the CoM-EN period (Figs 3 and 4). However, core area and distribution range re-expanded again during the Recovery period, until reaching dimensions similar to those observed during the MarZon period (Fig 3).

Unlike Puerto Ayora and Baquerizo Moreno, Puerto Villamil´s fishing fleet showed minimum variations of its core area and distribution range through the years, denoting a remarkable fidelity of Puerto Villamil's fishers to their traditional fishing grounds (Figs 3 and 4). Our results showed that spiny lobster fishing grounds located in the southern and western part of Isabela Island were used exclusively by fishers from Puerto Villamil. The only exception occurred during the reopening phase of the sea cucumber fishery (RovBan1), when Puerto Villamil and Puerto Ayora fishing fleets' core areas slightly overlapped (Figs 3 and 4). Puerto Villamil fishing fleet's distribution range showed a consecutive expansion and contraction pattern, similar to that described for Puerto Ayora during the same periods. However, unlike Puerto Ayora's fishers, who expanded their distribution range beyond Santa Cruz Island, Puerto Villamil's fishers remained fishing exclusively along the coastline of their home island, Isabela (Figs 3 and 4).

Hotspot analysis revealed the existence of significant fishing clusters across the archipelago and throughout the six periods analyzed (Figs 5 and 6). One of the the most relevant patterns identified by this analysis was that aggregation of fishing effort was not detected around the boundaries of no-take zones (Figs 5 and 6). Hotspots of fishing effort did not show large variations before and after the implementation of no-take zones (MarZon), suggesting that fishers were not displaced from their traditional fishing grounds nor attracted to no-take zone boundaries by a "fishing the line" effect.

According to port-based interviews, fishing effort showed high densities exclusively in the southern part of Isabela and San Cristobal Islands, near Baquerizo Moreno and Puerto Ayora, during CoM-En period (Fig 5). However, new hotspots appeared along the western and eastern parts of Santa Cruz Island, the western and southwestern parts of Isabela Island, the eastern part of San Cristobal Island and the southeastern part of Genovesa Island during RvBan1 and RVBan2 periods (Fig 5). Since then hotspot location patterns have shown few variations. Some hotspots disappeared sporadically during and after the sea cucumber overexploitation and marine zoning implementation phase, particularly those located in the western and southwestern part of Isabela Island (Fig 5). However, most hotspots have remained in the same locations through the years, particularly those located near fishing ports. Only during the spiny lobster recovery period, a single hotspot appeared in the southwestern part of Isabela Island, which had not been registered in previous periods, suggesting that fishing effort aggregated in new fishing grounds in 2011 (Fig 5).

Similar fishing patterns were identified by analysis of onboard observer data (Fig 6), although some slight variations were detected. According to this source of data, during the expansion phase of the sea cucumber fishery (RvBn2), there was a group of hotspots in the northwestern part of Marchena and Pinta Islands, as well as in the western and eastern part of

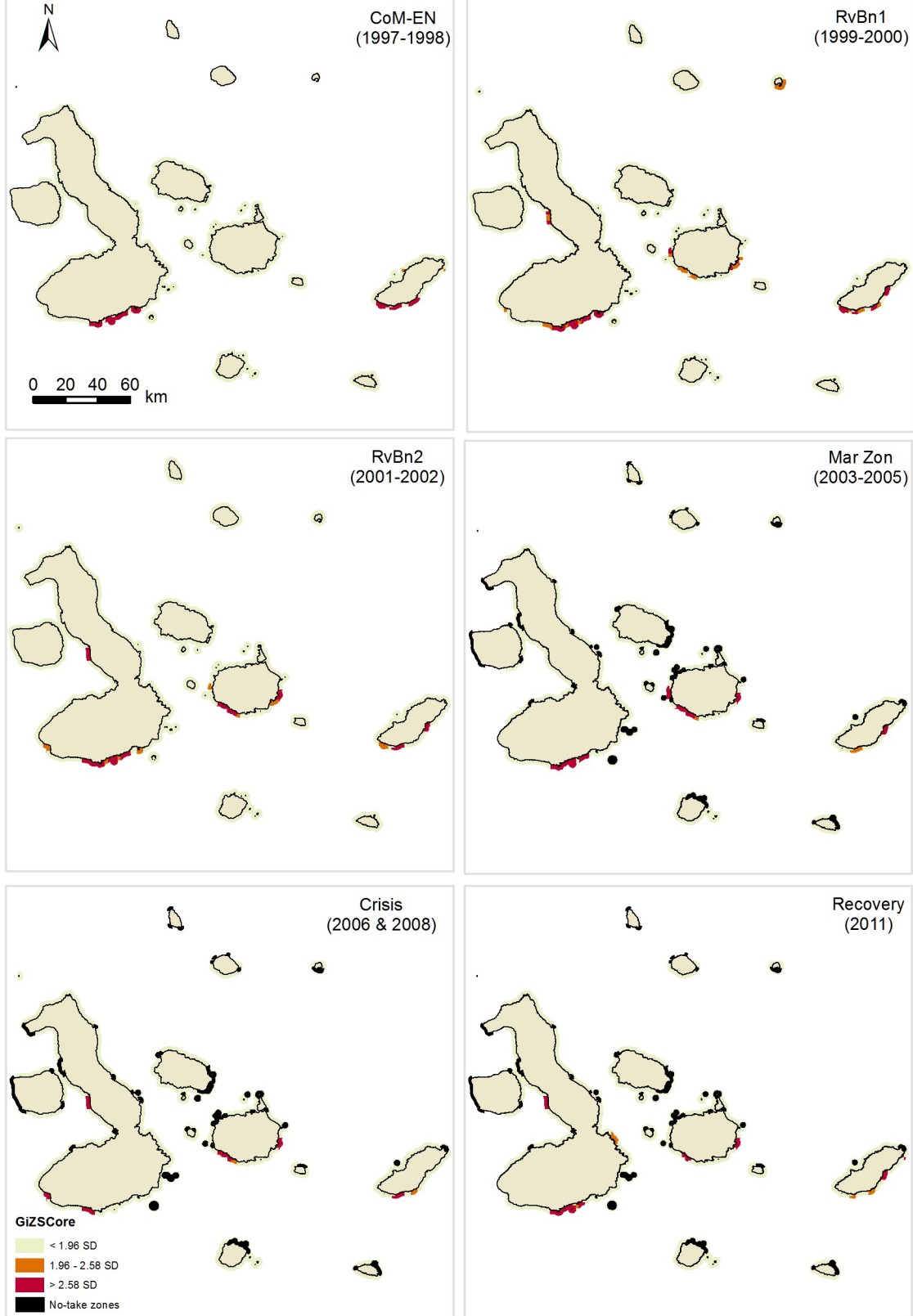

**Fig 5. Fishing effort hotspots in the Galapagos Marine Reserve for the spiny lobster fishery between 1997 and 2011, based on port interview data.** Six-time periods are shown: Co-governance and El Niño (CoM-EN); Sea cucumber re-opening phase

(RovBan1); Sea cucumber expansion phase (RovBan2); Sea cucumber overexploitation phase and marine zoning (MarZon); Sea cucumber collapse phase and global financial crisis (Crisis); and Spiny lobster recovery (Recovery).

Floreana Island, which were not detected by port-based interview data analysis. Despite these minor differences, the general fishing patterns identified by both sampling methods were quite similar (Figs 5 and 6).

### Climatic and human drivers affecting spatial fishing effort allocation

The BRT analysis was carried out with a focus on evaluating the relevance of no-take zones as a fishing effort predictor. However, NearNTZ was a relevant explanatory variable only for Puerto Villamil, and even in this case, the ranking of NearNTZ was very low (Table 4). This result suggests little if any effect of no-take areas on fishing patterns, at least for two of the three communities analyzed (Puerto Ayora and Baquerizo Moreno).

According to the regional BRT model, the most important fishing effort predictors were (Table 4): (1) DistHP, (2) PrevCatch, and (3) Period, followed by (4) latitude, (5) longitude, (6) SeaCucRev and (7) ONI. The remaining predictors were not useful in predicting fishing effort, as they performed worse than RN. Homeport BRT models showed, in most cases, similar patterns to the regional BRT model (Table 4). However, the contribution of fishing effort predictors varied among homeports, particularly in Baquerizo Moreno (Table 4). Based on the sum of the VI scores, static variables contributed very slightly more to the regional BRT model than dynamic predictor variables (Table 4, static: 50.0% dynamic: 45.3%). Likewise, the importance of static variables was higher than dynamic variables both for Puerto Villamil and Puerto Ayora BRT models (Table 4, PV: static (49.8%), dynamic (46.2%); PA: static (59.8%), dynamic (34.5%)). In contrast, dynamic predictor variables contributed more than static predictor variables in the BRT model for Baquerizo Moreno (Table 4, static: 37.0%; dynamic: 56.0%). According to the BRT model for Puerto Villamil, fishing effort was mostly influenced by eight variables, with DistHP, PrevCatch, latitude, and SeaCucRev, being the four most important predictors (Table 4).

The most important predictors contributing to the BRT model for Puerto Ayora were similar overall to Puerto Villamil, and DistHP was again the most important predictor. However, there were differences in other rankings (Table 4). Unlike Puerto Villamil, spiny lobster ex-vessel price performed better than RN in the BRT model for Puerto Ayora, while NearNTZ and SeaCucRev performed worse. The three most important predictors contributing to the BRT model for Baquerizo Moreno were, in contrast to Puerto Villamil and Puerto Ayora, longitude, ONI, and SeaCucRev (Table 4).

Performance statistics for the BRT models suggest that all models showed a good predictive performance to independent test data (Table 4). The regional BRT model explained 29.47% of the deviance in the data, while the Pearson's correlation coefficient was 0.55 (Table 4). Homeport BRT models showed, in most cases, better predictive performance than the regional BRT model (Table 4). Specifically, the BRT models for Puerto Villamil, Puerto Ayora and Baquerizo Moreno explained 35.73%, 32.66% and 15.74% of deviance in the data, respectively, while their Pearson's correlation coefficients were 0.59, 0.54 and 0.44, respectively.

Fishing effort was affected in different ways by the most influential variables identified by BRT models (Figs 7–10). Even though similar results were obtained by the regional and homeport BRT models, different patterns were observed among ports. All BRT models showed that fishing effort, measured as diver-hours, increased with distance to homeport, albeit with some variation (Figs 8–10). In Puerto Villamil, fishing effort increased gradually between 30 and 70 km from the homeport, leveling off subsequently (Fig 8). In contrast, for Puerto Ayora, fishing

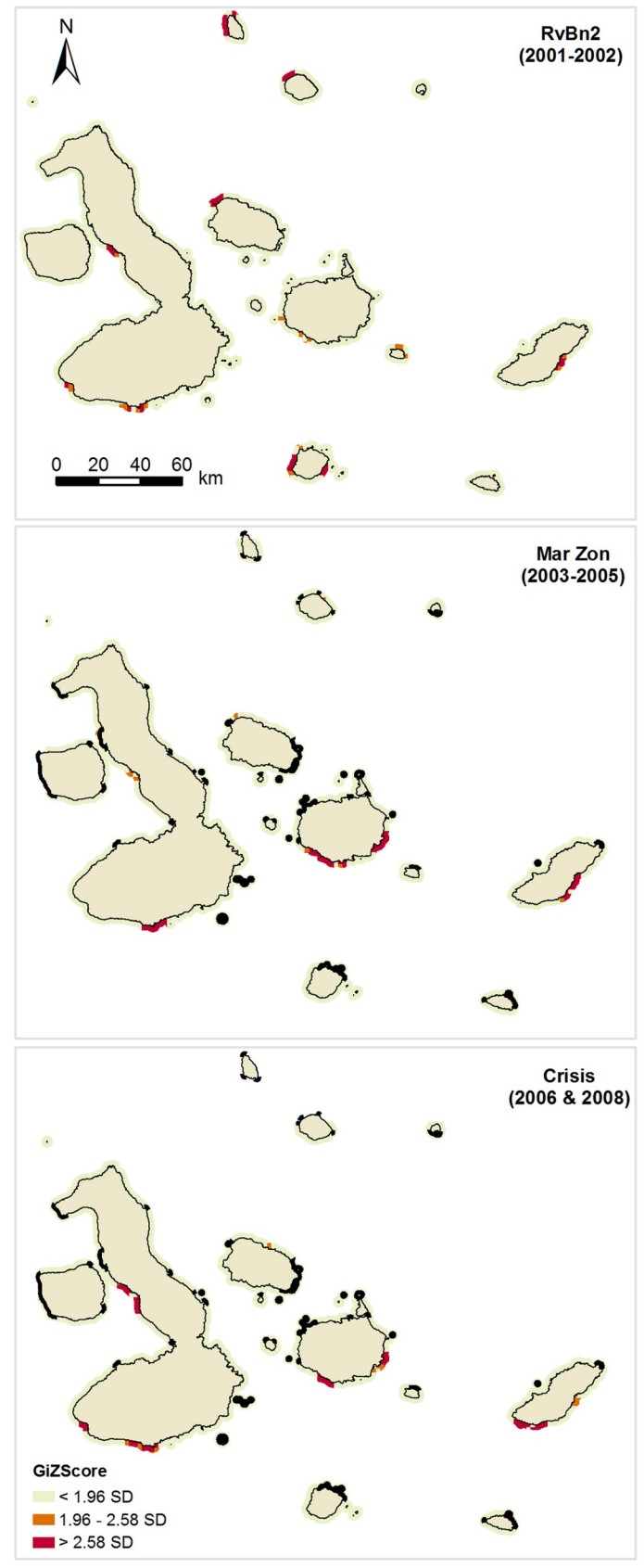

**Fig 6. Fishing effort hotspots in the Galapagos Marine Reserve for the spiny lobster fishery between 2001 and 2008, based on observer onboard data.** Three-time periods are shown: Sea cucumber expansion phase (RovBan2); Sea cucumber overexploitation phase and marine zoning (MarZon); Sea cucumber collapse phase and global financial crisis (Crisis).

effort increased gradually between 20 and 150km, then levelled off until 320 km, reaching maximum values after this distance (Fig 9). In other words, fishers from Puerto Ayora tend to fish farther away from their homeport in comparison with their peers from Puerto Villamil. In Baquerizo Moreno, while distance to homeport performed better than RN, its importance as a fishing effort predictor was much lower in comparison with Puerto Ayora and Puerto Villamil (Table 4; Fig 10).

Fishing effort showed a non-monotonic relationship with the previous spiny lobster catch in all BRT models. At regional level, fishing effort increased with previous catch until the latter reached 60 kg tail/fishing trip, then decreased until previous catch was approximately 105 kg tail/fishing trip, increasing again to a second peak at 110 kg tail/fishing trip, and leveling off subsequently. A similar pattern was observed at port level. In Puerto Villamil and Baquerizo Moreno, fishing effort also increased with the previous spiny lobster catch, to 30 and 55 kg tail/fishing trip, respectively, decreasing afterwards (Figs 8 and 10). In both cases, fishing effort increased again up to a previous spiny lobster catch at 105 and 60 kg tail/fishing trip, respectively, and leveled off subsequently. In contrast, fishing effort in Puerto Ayora increased abruptly up to a previous spiny lobster catch between 40 and 60 kg tail/fishing trip, decreasing afterwards (Fig 9).

**Table 4. For each predictor variable, the variable importance score (summing to 100) and the ranking is shown, for regional results and for each port.**

| Variable | Regional | Ranking | Puerto Villamil | Ranking | Puerto Ayora | Ranking | Baquerizo Moreno | Ranking |
|---|---|---|---|---|---|---|---|---|
| DistHP | **22.4** | **1** | **22.2** | **1** | **17.5** | **1** | 7.4 | 7 |
| Latitude | **9.0** | **4** | **10.5** | **3** | **15.7** | **3** | 8.4 | 6 |
| Longitude | **7.2** | **5** | **8.2** | **5** | **16.4** | **2** | 17.1 | 1 |
| Bioregion | 2.6 | 12 | 2.4 | 10 | 5.5 | 9 | 0.1 | 13 |
| NearNTZ | 4.7 | 9 | **6.5** | **7** | 4.7 | 10 | 4.0 | 10 |
| Period | **11.1** | **3** | **7.5** | **6** | **7.3** | **5** | 9.8 | 5 |
| Month | 1.9 | 13 | 1.3 | 12 | 2.1 | 11 | 5.7 | 9 |
| ExVesPrice | 4.1 | 10 | 2.0 | 11 | **6.2** | **7** | 2.3 | 11 |
| Vessel | 1.0 | 14 | 0.8 | 13 | 1.7 | 12 | 0.6 | 12 |
| ONI | **5.9** | **7** | **6.0** | **8** | **6.3** | **6** | 13.6 | 2 |
| SeaCucRev | **6.1** | **6** | **10.3** | **4** | 1.3 | 13 | 12.3 | 3 |
| PrevCatch | **15.2** | **2** | **18.3** | **2** | **9.6** | **4** | 11.7 | 4 |
| HomePort | 4.1 | 11 | NA | NA | NA | NA | NA | NA |
| RN | 4.8 | 8 | 4.0 | 9 | 5.8 | 8 | 7.0 | 8 |
| Sum of static variables importance | 50.0 | | 49.8 | | 59.8 | | 37.0 | |
| Sum of dynamic variables importance | 45.3 | | 46.2 | | 34.5 | | 56.0 | |
| Deviance explained (%) | 29.47 | | 35.73 | | 32.66 | | 15.74 | |
| Pearson's correlation coefficient (r) | 0.55 | | 0.59 | | 0.54 | | 0.44 | |

Bold numbers: predictor performed better than random numbers (RN). Shown at the bottom of the table are summary values for regional and homeport BRT models, i.e. the sums of static and dynamic VI scores, the deviance explained, and the Pearson's correlation coefficient. DistHP: distance to homeport; NearNTZ: distance to the nearest no-take zone; ExVesPrice: average ex-vessel price per year; PrevCatch: lobster catch obtained in previous fishing trip; SeaCucRev: average sea cucumber revenues obtained before the beginning of lobster fishing season season; ONI: Oceanic Niño Index; NA: No Applicable.

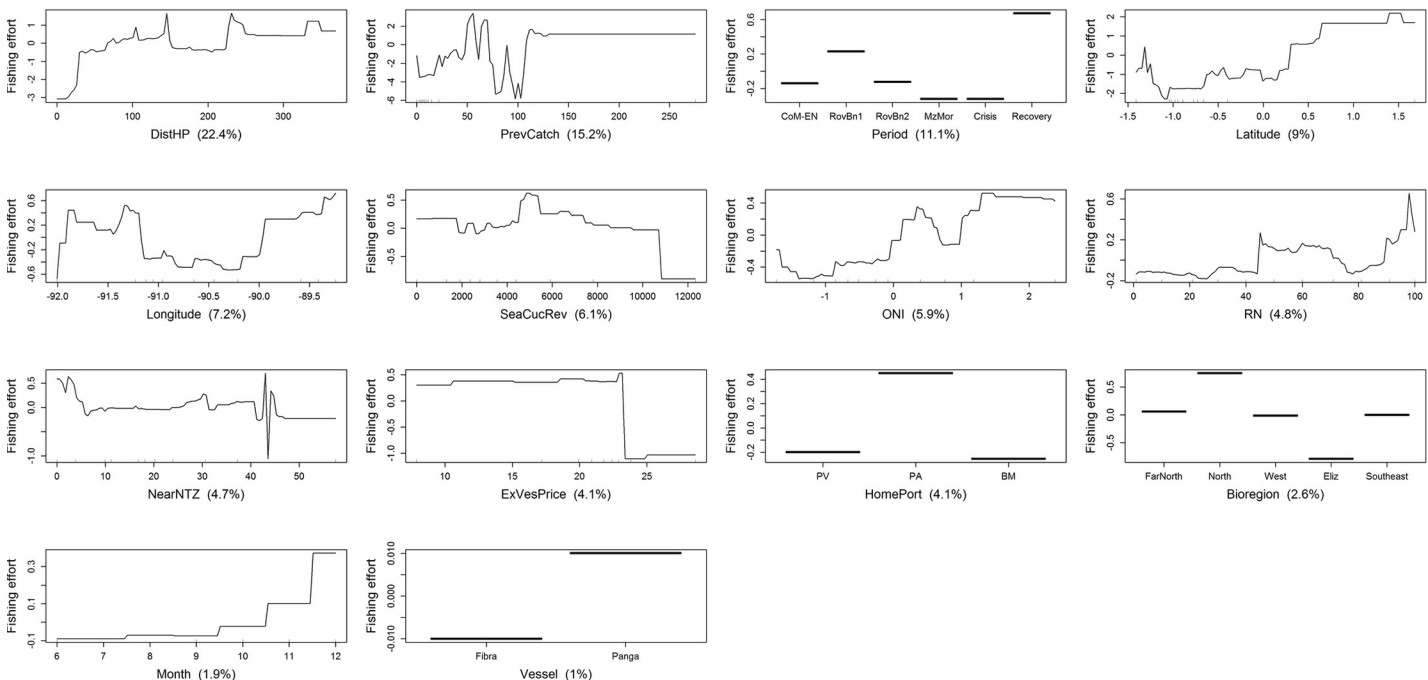

**Fig 7. Variation of fishing effort (in diver-hours) in relation to predictor variables for the spiny lobster fishery of the Galapagos Marine Reserve, according to the regional BRT model.** The response variable (diver-hours) has been centered by subtracting its mean. Variable importance scores are shown in parentheses. Rug plots indicate the distribution of observations in relation to the predictor variable. CoM-EN: Co-governance and El Niño; RovBn1: Sea cucumber re-opening phase; RovBn2: Sea cucumber expansion phase; MarZon: Sea cucumber overexploitation phase and marine zoning; Crisis: Sea cucumber collapse phase and global financial crisis; Recovery: Spiny lobster recovery.

Fishing effort showed different patterns according to the time period. At regional level, fishing effort increased dramatically during the reopening of the sea cucumber fishery (RovBN1), then declined gradually until reaching a minimum value during the sea cucumber collapse phase and global financial crisis 2007–09, increasing afterwards until reaching a maximum during the spiny lobster recovery period (Fig 7). A similar pattern was observed in Puerto Villamil, although in this case fishing effort decreased to a minimum during the marine zoning (MarZon) period, increasing afterwards (Fig 8). In Puerto Ayora, fishing effort also showed a maximum peak during RovBN1 period. However, unlike Puerto Villamil and Baquerizo Moreno, fishing effort decreased gradually until reaching a minimum value during the spiny lobster recovery period (Fig 9). In Baquerizo Moreno, fishing effort showed maximum values between CoM-EN and RovBN2 periods, then decreased until reaching a minimum value during the sea cucumber collapse phase and global financial crisis 2007–09. Afterwards, fishing effort increased slightly (Fig 10).

Spatially, fishing effort showed different distribution patterns across the archipelago, according to each fishing fleets' homeport. At regional level, fishing effort reached maximum values between 1.0˚ N and 1.5˚ N; i.e., around Darwin and Wolf islands, in the northernmost part of the Galapagos (Fig 7). In relation to longitude, fishing effort showed a decreasing trend from 91.3˚ W to 90.2˚ W; i.e., from the Eastern side of Fernandina toward Santiago Island; this was then followed by a steep increase from 90.0˚ W to 89.5˚ W; i.e., from the Western part of Santa Cruz Island toward Española and the Southwestern part of San Cristobal. Partial dependence plots for homeports showed that Puerto Ayora's fishing fleet was the only one that fished around these two islands between 1997 and 2011 (Fig 9). In this case, fishing effort showed a decreasing trend latitudinally from 90.0˚ W to 92.0˚ W; i.e., from the Western of Santa Fe

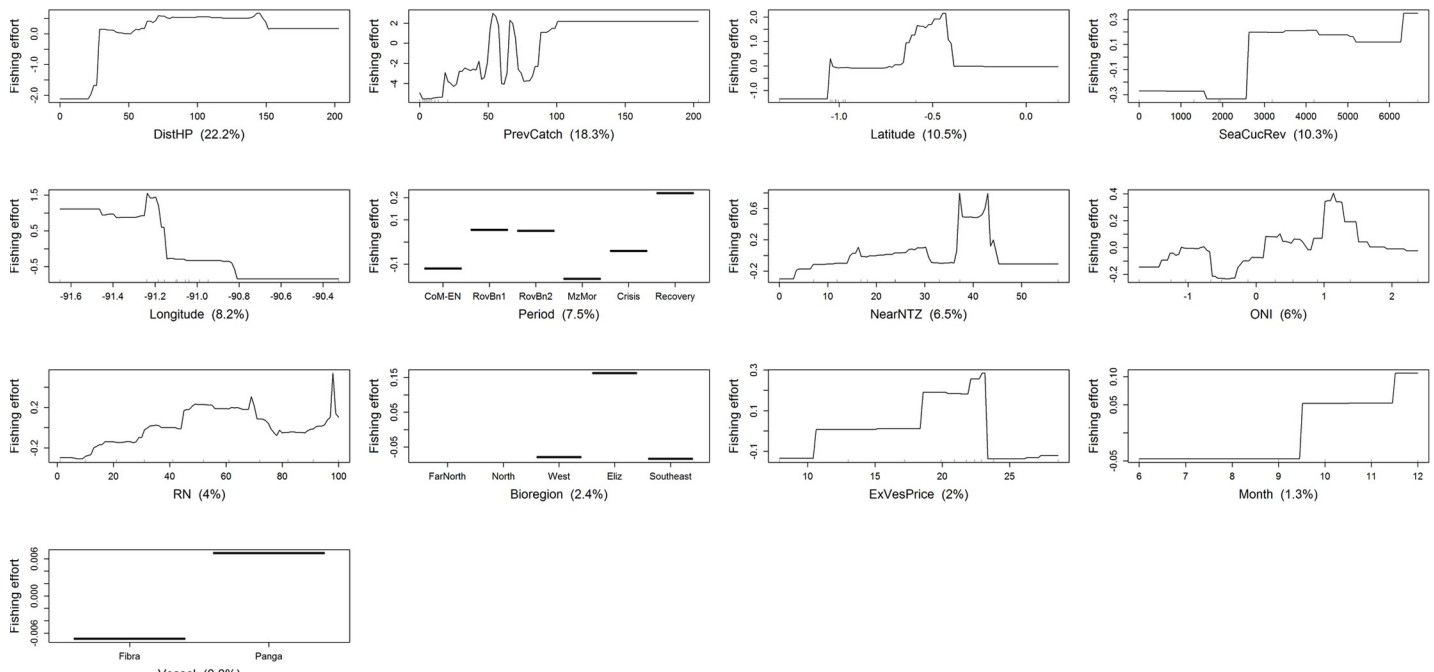

**Fig 8. Variation of fishing effort (in diver-hours) in relation to predictor variables for the spiny lobster fishery of the Galapagos Marine Reserve, according to the BRT model for Puerto Villamil.** The response variable (diver-hours) has been centered by subtracting its mean. Variable importance scores are shown in parentheses. Rug plots indicate the distribution of observations in relation to the predictor variable. CoM-EN: Co-governance and El Niño; RovBn1: Sea cucumber re-opening phase; RovBn2: Sea cucumber expansion phase; MarZon: Sea cucumber overexploitation phase and marine zoning; Crisis: Sea cucumber collapse phase and global financial crisis; Recovery: Spiny lobster recovery.

Island toward Santa Cruz, Isabela and Fernandina Islands. In contrast, fishing effort for Puerto Villamil showed two peaks at latitudes 1.1 S and 0.5 S, while longitude showed an increasing trend in fishing effort from 90.8˚ W to 91.2˚ W (Fig 8). These results suggest that fishing effort increased from Puerto Villamil toward the Southwestern part of Isabela Island, reaching a peak probably in the hotspot located in the South part of this island (Figs 5 and 6). Then fishing effort increased toward the North, reaching a peak probably in the hotspot located in the Western side of Isabela Island (Figs 5 and 6). Finally, fishing effort for Baquerizo Moreno showed a peak at -1.5˚S, probably around Española Island, then increasing very slightly from South to North (Fig 10). These results suggest that fishing effort is largely influenced by the location of fishing ports, as it was also demonstrated by the strong performance of the DistHP variable as a fishing effort predictor (Table 4) and by the location of hotspots and fishing fleets' core areas in fishing grounds adjacents to Puerto Villamil, Puerto Ayora and Baquerizo Moreno (Figs 3–6).

Most BRT models showed that fishing effort increased when sea cucumber revenues, obtained the fishing season before the beginning of lobster season (SeaCucRev), ranged between US$2000 and US$6,400/fishing season. At regional level, fishing effort decreased with higher sea cucumber revenues, including a steep decrease when revenues exceeded US$11000/ fishing season (Fig 7). In Puerto Villamil, fishing effort increased with sea cucumber revenues up to the latter reaching US$2500/fishing season, showing a second and highest peak when revenues were higher than US$6400/fishing season (Fig 8). In Baquerizo Moreno, fishing effort showed a positive relationship with sea cucumber revenues only when they reached values between US$4000 and US$5000/fishing season (Fig 10). Before this threshold no relationship was found. In Puerto Ayora, this explanatory variable was not relevant (Fig 9).

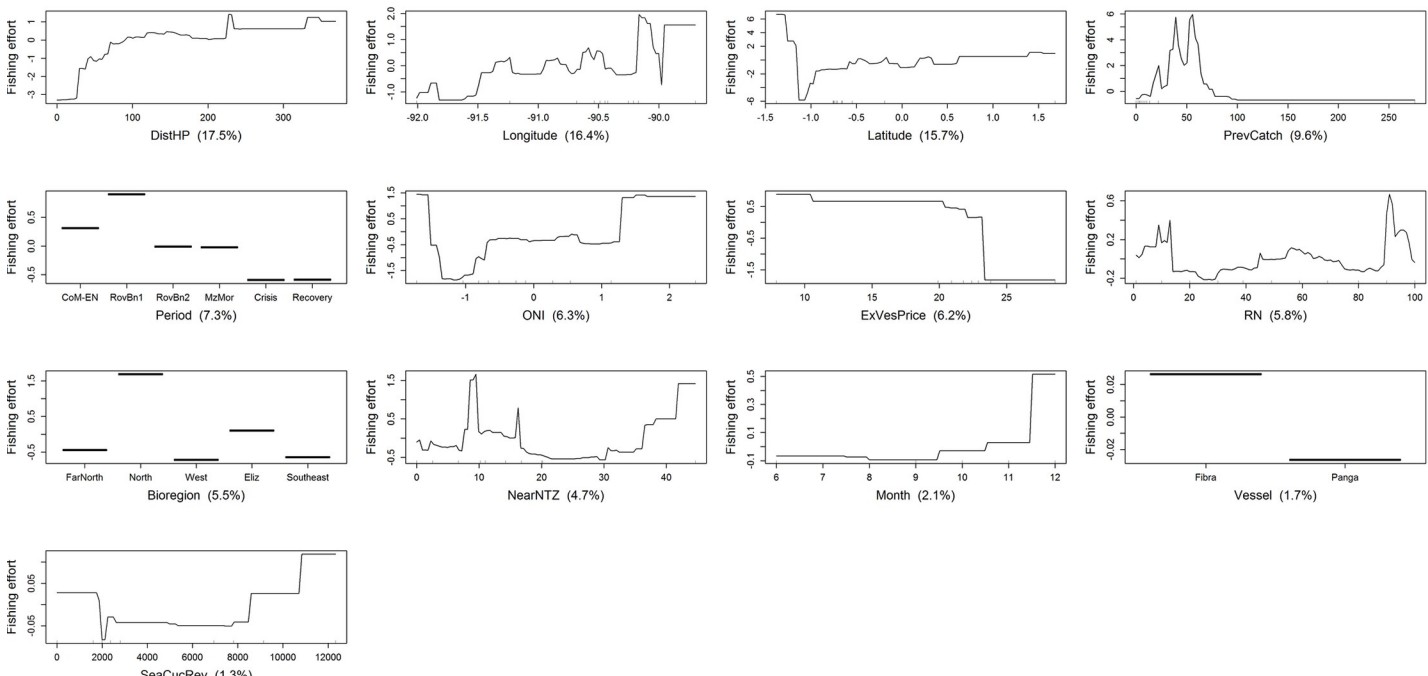

**Fig 9. Variation of fishing effort (in diver-hours) in relation to predictor variables for the spiny lobster fishery of the Galapagos Marine Reserve, according to the BRT model for Puerto Ayora.** The response variable (diver-hours) has been centered by subtracting its mean. Variable importance scores are shown in parentheses. Rug plots indicate the distribution of observations in relation to the predictor variable. CoM-EN: Co-governance and El Niño; RovBn1: Sea cucumber re-opening phase; RovBn2: Sea cucumber expansion phase; MarZon: Sea cucumber overexploitation phase and marine zoning; Crisis: Sea cucumber collapse phase and global financial crisis; Recovery: Spiny lobster recovery.

Finally, at regional level, fishing effort increased when the oceanographic variable ONI ranged between -1.0 and 2.0. Similar patterns were observed among ports (Figs 8–10). In Puerto Villamil and Puerto Ayora, fishing effort showed an increasing trend from -1.0 to 1.5 (Figs 8 and 9), while in Baquerizo Moreno fishing effort increased gradually after 0, showing a peak around 2.0 (Fig 10). ONI values equal to or higher than +0.5 indicate El Niño conditions, meaning that the East-central tropical Pacific is significantly warmer than usual. In contrast, ONI values equal to or lower than -0.5 indicate La Niña conditions, meaning that the region is cooler than usual. Therefore, our results suggest that fishing effort increased during El Niño conditions.

## Discussion

To our knowledge, this paper represents the first empirical study that illustrates how GIS techniques can be used in combination with BRT models to evaluate and predict the spatial distribution of fishing effort, and its response to human and climatic drivers of change, in this case, inside a multiple-use MPA. Our results showed that substantial changes in the spatio-temporal distribution of fishing effort occurred in the Galapagos spiny lobster fishery due to interaction among various climatic and human drivers, acting at multiple temporal and spatial scales. In this section, we first reviewed how these drivers of change influenced the large-scale dynamics of fishing effort in the Galapagos spiny lobster fishery. That is followed with an examination of micro-scale dynamics of fishing patterns in the small-scale fleet. Finally, we examined the impact of no-take zones and their management implications, leading to recommendations to

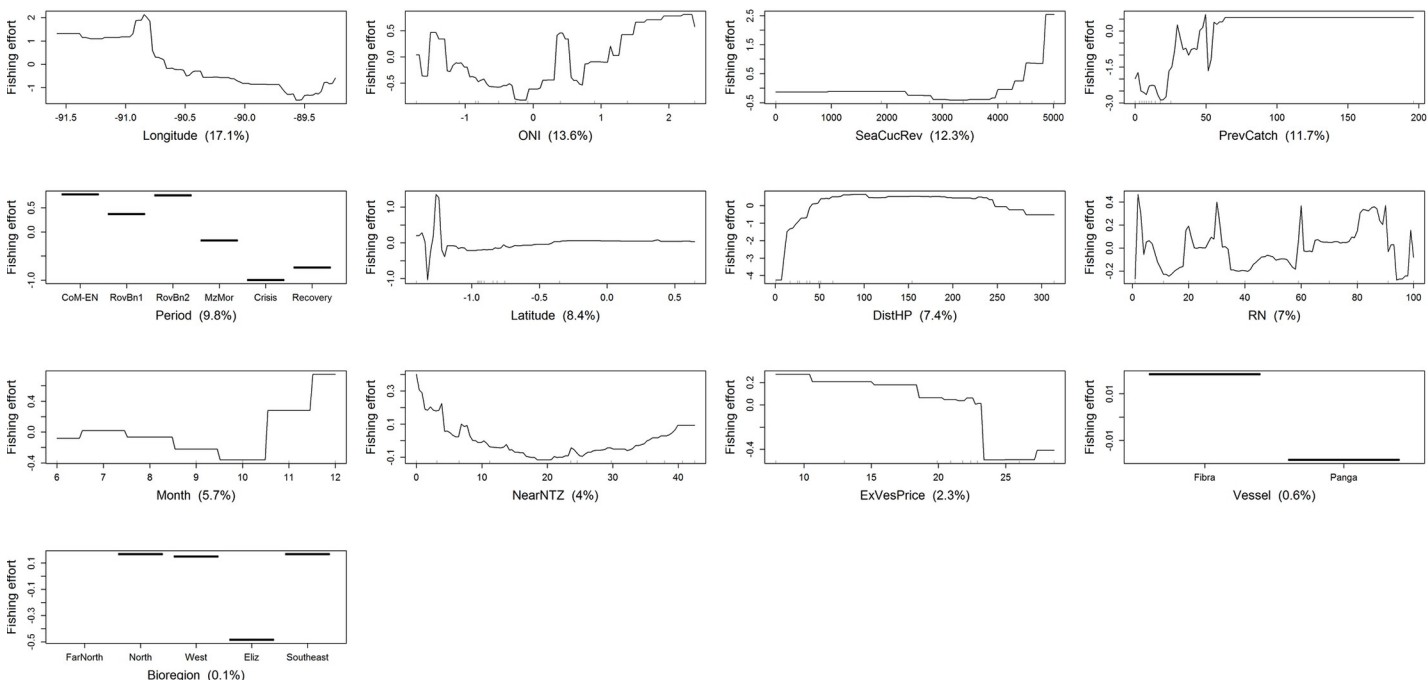

**Fig 10. Variation of fishing effort (in diver-hours) in relation to predictor variables for the spiny lobster fishery of the Galapagos Marine Reserve, according to the BRT model for Baquerizo Moreno.** The response variable (diver-hours) has been centered by subtracting its mean. Variable importance scores are shown in parentheses. Rug plots indicate the distribution of observations in relation to the predictor variable. CoM-EN: Co-governance and El Niño; RovBan1: Sea cucumber re-opening phase; RovBan2: Sea cucumber expansion phase; MarZon: Sea cucumber overexploitation phase and marine zoning; Crisis: Sea cucumber collapse phase and global financial crisis; Recovery: Spiny lobster recovery.

improve the design and effectiveness of Galapagos marine zoning to reconcile conservation and fishery management objectives.

### Drivers of change: 'Macro' adaptive responses

Our results showed that the spiny lobster fishery of the Galapagos, and the spatio-temporal distribution of its fishing effort, were especially affected by two major drivers, the boom-and-bust exploitation of the sea cucumber fishery and the global financial crisis 2007–09. Both drivers of change triggered substantial macro-scale changes in fishing effort dynamics within the GMR, notably reflected in a remarkable reduction of fishing capacity, and shifts in post-harvest arrangements.

The sequence of events examined here began with the re-opening of the sea cucumber fishery, one year after the creation of the GMR. This event caused severe overcapitalization of the entire Galapagos small-scale fishing sector, with fishing capacity increasing not only in the sea cucumber fishery but also in the lobster fishery [2,24,52]. Only a moratorium on new entrants in 2002 stopped the exponential growth in the number of fishers and vessels registered in Galapagos, that had occurred between 1997 and 2000 [51]. Subsequently, the collapse of the sea cucumber fishery in 2006 caused a severe economic perturbation, which was intensified a few years later by the global financial crisis 2007–09 [38,52]. The latter led to a sharp contraction in the consumption of lobsters in the United States, the main foreign market for Galapagos lobsters [53], and a price drop of 32% between 2008 and 2009 [53]. In response to this economic perturbation, a significant number of fishers abandoned not only the sea cucumber fishery, but also the spiny lobster fishery, leading to a 56% reduction in fishing effort between 2005

and 2008 [2]. This resulted in declines of total catch, and exports to mainland Ecuador, by 23% and 45%, respectively [2,38].

Our results suggest that those fishers who decided to remain in the spiny lobster fishery after 2006 responded to the crisis either by re-expanding their distribution ranges and core areas (in the cases of Puerto Villamil and Baquerizo Moreno) or by diversifying their products and markets (for Puerto Ayora). In the latter case, some fishers reacted to the global financial crisis 2007–09 in two ways [38]: (1) diversifying their product by trading whole fresh lobsters instead of lobster tails, as had been done since the 1960s; and (2) diversifying their market by selling their product directly to the local hospitality sector and general public instead of middlemen. The restructuring of the value chain improved fishers' revenues by increasing local consumption of whole lobsters and increasing ex-vessel prices [54]. Diversification of products and markets was enabled by the fact that tourist and land-based infrastructure (hotels and restaurants) is more extensive in Puerto Ayora than in Baquerizo Moreno or Puerto Villamil. This socioeconomic feature made fishers from Puerto Ayora less vulnerable to the economic perturbations caused by total closure of the sea cucumber fishery and the global financial crisis 2007–09. Puerto Ayora's fishers faced the crisis by adding value to their catches, rather than expanding the spatial range of fishing, as did fishers in other ports. Indeed the Puerto Ayora fishers shifted their fishing effort to nearer their homeport, which likely increased their profits by reducing variable costs (e.g., diesel fuel).

Different adaptive responses to the global financial crisis 2007–09 were reported by Castrejón and Defeo [38] for two Mexican spiny lobster fisheries, one in Punta Allen, Quintana Roo, and the other in Baja California. In Punta Allen, fishers from Vigía Chico's fishing cooperative stopped lobster fishing for three months until market conditions improved and, since then, they have acquired the infrastructure, technology and expertise needed to export live lobsters to Asia and Europe. In Baja California, fishers from the Federation of Cooperative Societies of the Fishing Industry of Baja California (FEDECOOP) also adapted their harvesting and trading strategies according to global market conditions. In this case, a 10-day early closure of the spiny lobster fishing season was agreed upon, and implemented in a coordinated way, by the 10 cooperatives that made up the FEDECOOP. Thanks to this, and other harvest and trading strategies, such as the agreement of spiny lobster unit price and harvesting levels before the beginning of each fishing season, and the diversification of markets and products, FEDECOOP was able to reach maximum historic prices after the conclusion of the global financial crisis 2007–09. These two case studies, together with the Galapagos spiny lobster fishery, reinforce the notion that crises represent opportunities for learning, adapting, and entering onto more sustainable pathways [55]. Such crises triggered adaptive responses, either individual or collective, which were shaped by the social and geographic attributes of the fishing communities in which fishing cooperatives are embedded, and the capacity and willingness of individuals and fishing cooperatives to take actions to re-organize themselves, change harvesting and trading strategies, and implement self-regulatory mechanisms to face the economic perturbations caused by external drivers of change [38].

Finally, it should be noted that while the global financial crisis 2007–09 was detrimental for Galapagos fishers, it was beneficial for spiny lobster stocks. Two years after the official end of the recession, lobster CPUE and catch increased 91% and 102%, respectively, whereas fishing effort only increased 6% between 2009 and 2011 [2]. According to Defeo et al. [56], the recovery of spiny lobster stocks could be attributed to the substantial reduction in fishing effort, together with the combined effect of market forces and favorable environmental conditions. Our results support this hypothesis.

## Drivers of change: 'Micro' adaptive responses

The macro-scale changes in the spiny lobster fishery, described above, in turn altered the micro-scale dynamics of fishing patterns in the small-scale fleet, reflected in spatio-temporal variations in the fishing fleets' core areas and fishing effort distribution. According to the BRT models, these changes in fishing effort were shaped by six main predictor variables: the distance from homeport (DistHP), the latitude and the longitude, the particular time period considered (Period), the Oceanic Niño Index (ONI), and the lobster catch obtained in previous fishing trips (PrevCatch). The average sea cucumber revenue obtained the fishing season before the beginning of lobster season (SeaCucRev) also showed a good predictive performance, but not for Puerto Ayora, for the reasons explained above.

The coastal nature of the spiny lobster fishery, and the geographic and socioeconomic features of each homeport, help to explain why the six explanatory variables were the most relevant as fishing effort predictors. Each port showed different adaptive responses to these drivers due to differences regarding number of fishers, composition of the fishing fleet, and available land-based tourism infrastructure. For example, Baquerizo Moreno has historically had the largest concentration of fishers and mother boat vessels [51]. Such features probably have forced fishers to fish farther away from their homeport to reduce competition with their peers, thereby reducing the influence of static variables on fishing effort distribution. In contrast, the reduced number of mother boats in Puerto Ayora and Villamil increased the influence of static variables, which may explain why these fishers catch spiny lobsters near their homeports.

A special feature of Galapagos is the limited number of landing sites (Fig 1). This, together with the limited range of the Galapagos small-scale fishing fleet and the close proximity of homeports to the most productive fishing grounds [36,53], explains why static rather than dynamic explanatory variables were more relevant as fishing effort predictors. Thus, distance to homeport, latitude and longitude are the most important variables explaining why fishers from the same homeport tend to use similar fishing grounds. This leads to exclusive core areas, which usually do not show overlaps. Similar results were found by Bucaram et al. [20] whose short-term analysis of factors affecting fishing behaviour in the Galapagos spiny lobster fishery identified travel distance from vessels' home ports to fishing grounds and expected revenues as the most important factors affecting spatial allocation of fishing effort. They also found that fishing behaviour is sensitive to changes in sea conditions and sea surface temperature, but not to precipitation or moon visibility.

In contrast, dynamic variables explain why fishing patterns changed during certain periods of time. In this sense, our results suggest that two variables, the revenues produced by the sea cucumber fishery and the previous lobster catch, are responsible for the alternate expansion and contraction of fishing fleets' core areas and distribution ranges during the boom-and-bust exploitation of the sea cucumber fishery, marine zoning implementation and the global financial crisis 2007–09. Higher revenues produced by the sea cucumber fishery and higher previous lobster catches were associated with increasing trends in fishing effort in the spiny lobster fishery. Our results suggest that higher revenues produced during the reopening and expansion period of the sea cucumber fishery probably acted as subsidies that allowed spiny lobster fishers to extend their fishing trips for longer times and farther away from their homeports. In contrast, lower revenues caused by the overexploitation of the sea cucumber and spiny lobster fishery, and the global financial crisis 2007–09, led to decreasing trends in fishing effort, which were reflected in the contraction of fishing fleets' core areas and distribution ranges. These hypotheses are supported by Bucaram and Hearn [57], who found that the decision to participate in the spiny lobster fishery is significantly influenced by the average catch per trip of spiny lobsters and sea cucumbers during the previous fishing season. In other words, the higher the

average catch per trip of both species obtained by a vessel in the previous year, the more likely that the vessel will decide to participate in the next spiny lobster fishing season, and the greater the extent of participation.

The oceanographic variable ONI was also identified in our analysis as a relevant fishing effort predictor. From partial dependence plots, fishing effort increased during El Niño conditions; this could be caused by the redistribution of spiny lobster stocks from inshore to deeper waters, making them inaccessible to fishing by hooka diving (cf. [58]). Such reproductive migrations are influenced by temperature. According to Vega [59], warmer temperatures during El Niño periods accelerate the time of breeding of *Panulirus interruptus* significantly, while the converse occurred under colder temperatures caused by La Niña. Based on these studies, fishing effort probably was higher in the Galapagos spiny lobster fishery during El Niño events, with reproductive migration of spiny lobsters to deeper waters making the lobster less accessible to fishing, leading to increased search times and increased diving hours per fishing trip.

## Impact of no-take zones and management implications

The above results of our integrated analysis showed how Galapagos fishing communities coped with the interactions of human and climatic drivers of change, both temporally and spatially. Our results showed that 'macro' and the 'micro' fishers' adaptive responses varied according to the magnitude, extent and intensity of the social-ecological perturbations caused by the drivers of change analyzed, and were shaped by geographic, economic and oceanographic factors, including the socioeconomic attributes of the three fishing communities analyzed.

Furthermore, our results indicated that among the possible effects producing a recovery of the spiny lobster stock, the implementation of no-take zones within the GMR was not a significant factor. Indeed, there is no scientific evidence (see also [2]) that adoption of no-take zones contributed directly to the sustainability of Galapagos shellfish fisheries [60,61]. The lack of a 'fishing the line' effect around no-take zones and the poor performance of the NearNTZ variable as a fishing effort predictor suggest that marine zoning, after its implementation, had little impact on the spatio-temporal distribution of fishing effort, particularly in Puerto Ayora and Baquerizo Moreno. This result may well be due to the manner in which locations of no-take zones were chosen across Galapagos. According to Edgar et al. [21], fishers sought to minimize perceived impacts on their livelihood by advocating the location of no-take zones in areas with low densities of the most valuable commercial species (sea cucumber and spiny lobster), while tourism operators and sport divers promoted the protection of areas containing high densities of species important for tourism (e.g., sharks). As a result, sea cucumber and spiny lobster baseline densities were much more abundant (3 and 2.7 times higher, respectively) in fishing zones compared to no-take zones, although differences between both zone types were not significant for spiny lobsters [21]. The location of no-take zones in areas with relatively low abundance of the most lucrative species explains why fishers have shown a lack of interest in fishing near these areas, thereby explaining the lack of a 'fishing the line' effect around no-take zone boundaries after marine zoning implementation.

The above results have implications for fisheries policy and management in the GMR. First, there is a need to re-evaluate the distribution of no-take zones across the GMR, to promote the sustainability of the spiny lobster fishery and conserve key biodiversity areas. Effective no-take zones should be implemented in areas that ensure the protection of a proportion of the breeding stock and critical reproduction and nursery habitats. However, the geographic location of these areas across the Galapagos archipelago is still uncertain. It may be useful to consider fishing effort hotspots, which probably overlap with the location of spawning and nursery areas, a

hypothesis that should be evaluated by future studies. Unfortunately, declaring fishing effort hotspots as no-take zones will represent a challenge, considering the lack of evidence of the ecological and economic benefits provided by no-take zones and the high opportunity and transaction costs associated with their implementation and enforcement. That reality has contributed to reduce the acceptability and legitimacy of what could be potentially a valuable tool to manage Galapagos shellfish fisheries [22]. In consequence, additional research and management efforts are required to create the conditions for the effective planning, implementation, monitoring and enforcement of the GMR's marine zoning.

Second, since no-take zones represent only one of multiple management tools available for successful implementation of spatial EBM [22], a more effective approach could be a combination of a coastal network of no-take zones with co-managed harvested areas that allocate exclusive spatial fishing rights to local communities, potentially a more robust approach to address the roots of fisheries management failures that led to overexploitation of fisheries [2,17,18,62]. This could produce a set of strategically-placed Territorial Use Rights in Fishing (TURF) areas. Based on the distribution of core areas and fishing effort hotspots across the archipelago for the three fishing fleets, the most strategic places for the experimental implementation of TURFs in the GMR are those located in the southern part of Isabela Island, the western part of Santa Cruz Island and the southeastern part of San Cristobal Island. These places are strategic because (1) each is used exclusively by one fishing fleet, which reduces the likelihood of potential conflicts among different fishing fleets arising over competition for the same ocean space, and (2) each is located near the corresponding homeport, which facilitates surveillance, control and monitoring activities and creates an economic incentive for TURF co-management. The active involvement of local communities in the co-management of strategically placed TURFs could contribute, under certain enabling conditions, to generate a sense of stewardship among fishers [4,63–65]. This management approach could promote the implementation, by fishers themselves, of effective monitoring, control and surveillance procedures, and the accomplishment of objectives for management and conservation [66], as has been observed in spiny lobster fisheries of Baja California, México and Chile [38,67,68], all currently certified by the Marine Stewardship Council as sustainable [69].

Third, the geographic definition of new management areas is needed, based on core areas and distribution ranges of the three fishing fleets. This could include area-based co-management that could enhance the acceptability and legitimacy of GMR's marine zoning and help to mitigate the potential conflict associated with the redistribution of no-take zones. Within the area-based co-management system, we suggest creating specific co-management councils for each management area to promote the involvement and participation of local stakeholders in their planning, implementation, monitoring and enforcement. Each co-management council should be made up exclusively of those fishers, and other relevant stakeholders, who would be most affected by implementation of no-take zones, and/or the experimental allocation of TURFs, inside their core area and distribution ranges. An area-based co-management approach could be useful to ensure a strategic distribution of no-take zones across the archipelago and to minimize the impact of zoning on fishing communities' livelihoods, helping to improve the acceptance, legitimacy and compliance of the new marine zoning scheme.

## Conclusions and lessons learned

The first and most important conclusion and lesson learned lies in the reality that fishery systems face social-ecological impacts produced by a diverse range of human and climatic external drivers of change, acting at different spatial and temporal scales, usually simultaneously. These include not only the implementation of new regulations, such as multiple-use MPAs

and marine zoning schemes, and extreme climatic events associated with global climate change (e.g., El Niño), but also socioeconomic perturbations caused by the globalization of markets and the development and/or collapse of alternative fisheries.

Second, our results demonstrated the need for a broad-based and integrated social-ecological approach to fishery management and marine conservation, whether in planning MPAs or in any fisheries management, or indeed, any natural resource management. In this context, MPAs must be designed and implemented taking into consideration not only the spatial-temporal dynamics of key biodiversity areas and fishery resources, but also the dynamics of fishing fleets, and fishers' adaptive responses to human and climatic drivers of change. Only in this way will management and conservation measures, such as no-take zones, be useful tools for rebuilding depleted fish stocks, conserving marine ecosystems and improving fishing communities' livelihoods.

Third, assessments of the effectiveness of MPAs should not assume that any change in fishing patterns is caused exclusively by the implementation of an MPA. Instead, a comprehensive understanding of how local fishing communities cope with relevant human and climatic drivers is fundamental to properly assess the socio-ecological outcomes generated by an MPA. This knowledge will reduce the risk of errors in planning, implementing and assessing the effectiveness of MPAs and marine zoning more broadly.

These conclusions and lessons learned apply broadly, not only to situations involving MPAs, but to any social-ecological system in which ecosystem-based management, marine zoning and other management approaches are being considered to improve the governance and sustainability of fisheries and the conservation of key biodiversity areas.

## Supporting information

**S1 Table. Summary of the fishery monitoring data gathered for the spiny lobster fishery at the three main ports of the Galapagos Marine Reserve from 1997 to 2011.**
(DOCX)

**S2 Table. Fishing fleets estimated mean core areas and distribution ranges (in km$^2$), according to port interviews and observer onboard data collected in the Galapagos Marine Reserve from 1997 to 2011.**
(DOCX)

**S3 Table. Fishing fleets estimated site fidelity (IOR95) to similar core areas and distribution ranges, according to port interviews and observer onboard data collected in the Galapagos Marine Reserve from 1997 to 2011.**
(DOCX)

## Acknowledgments

We thank Peter Tyedmers, Boris Worm, Jeffrey Hutchings, Robert Steneck, as well as the editor and referees, for their helpful suggestions and comments to improve this research article. We acknowledge the Galapagos National Park Service and the Charles Darwin Foundation for sharing the fishery-related data required to conduct this study.

## Author Contributions

**Conceptualization:** Mauricio Castrejón, Anthony Charles.

**Data curation:** Mauricio Castrejón.

**Formal analysis:** Mauricio Castrejón.

**Funding acquisition:** Mauricio Castrejón, Anthony Charles.

**Investigation:** Mauricio Castrejón, Anthony Charles.

**Methodology:** Mauricio Castrejón.

**Supervision:** Anthony Charles.

**Writing – original draft:** Mauricio Castrejón.

**Writing – review & editing:** Mauricio Castrejón, Anthony Charles.

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
