## [Decision Letter · Decision Letter 0]

1 Nov 2019

PONE-D-19-26580

Human and climatic drivers affect spatial fishing patterns in a multiple-use marine protected area: the Galapagos Marine Reserve

PLOS ONE

Dear Dr. Castrejón,

Thank you for submitting your manuscript to PLOS ONE. After careful consideration, we feel that it has merit but does not fully meet PLOS ONE’s publication criteria as it currently stands. Therefore, we invite you to submit a revised version of the manuscript that addresses the points raised during the review process.

I found this to be a really interesting study. The inclusion of social drivers was especially interesting. However, both reviewers have flagged a number of issues that need to be addressed before the manuscript can be published.  The main issue with this manuscript is that it is just far too long and wordy. It would be much more accessible for the reader if it was written in a more concise manner. Instead, it is very repetitive and often includes details that are not germane to the actual study. The Discussion is especially repetitive and in need of some serious editing. Both reviewers have provided very detailed comments to improve the clarity of the paper and assist the authors in their revisions. I have also provided extensive editorial comments (PLoS editorial comment file) to assist the authors. I strongly encourage the authors to consider all the comments provided when making your revisions.

We would appreciate receiving your revised manuscript by Dec 16 2019 11:59PM. To enhance the reproducibility of your results, we recommend that if applicable you deposit your laboratory protocols in protocols.io, where a protocol can be assigned its own identifier (DOI) such that it can be cited independently in the future. For instructions see: http://journals.plos.org/plosone/s/submission-guidelines#loc-laboratory-protocols

We look forward to receiving your revised manuscript.

Kind regards,

Heather M. Patterson, Ph.D.

Academic Editor

PLOS ONE

Journal Requirements:

Reviewers' comments:

Reviewer's Responses to Questions

**Comments to the Author**

1. Is the manuscript technically sound, and do the data support the conclusions?

Reviewer #1: Yes

Reviewer #2: Yes

2. Has the statistical analysis been performed appropriately and rigorously? 

Reviewer #1: Yes

Reviewer #2: Yes

3. Have the authors made all data underlying the findings in their manuscript fully available?

Reviewer #1: Yes

Reviewer #2: Yes

4. Is the manuscript presented in an intelligible fashion and written in standard English?

Reviewer #1: Yes

Reviewer #2: Yes

5. Review Comments to the Author

Reviewer #1: Review of PONE-D-19-26580. “Human and climatic drivers affect spatial fishing patterns in a multiple-use marine protected area: the Galapagos Marine Reserve”. Castrejon, M. and A. Charles

This study evaluated how the spatiotemporal allocation of fishing effort for lobsters in the Galapagos multiple-use Marine Protected Area was affected by the interaction of diverse climatic and human drivers, before and after implementation of no-take zones. The study used GIS data on fishing effort and BRTs to attempt to identify how these drivers affected spatial fishing patterns. The paper concludes that the boom-and-bust exploitation of the sea cucumber fishery and the global financial crisis (2007-2009), rather than no-take zone implementation, were the most important drivers affecting the distribution of fishing effort for lobsters across the archipelago. The study is spatially and temporally extensive (most of the Galapagos Islands, 1997-2011), the data are fairly well-analyzed and interpreted, and the manuscript is well-written. I have no major disagreements with the conclusions. I also have some sympathy with the suggestions that the MPA network placement could be revisited or even revised, and outside the network TURFS encouraged. My comments are mostly to assist the authors with publication.

Major Comments.

1. This is a paper that, in effect, quantifies spatial and temporal trends in fishing effort of a lobster fishery in a developing country. Yet the emphasis chosen is how this data informs effects of MPA implementation. It is highly commendable that the study includes before and after implementation data. In fact, this is such an important aspect of the study, I would recommend that the authors stress this point more in the paper. However, this MPA network is also well-known as a “classic” case where fishers ensured that no-take zones were NOT placed where fishers fished (Edgar et al 2004 Ref. 22 in this manuscript). That is, it is a case where you might NOT expect much change in spatial effort in the lobster fishery pre- and post-implementation of the MPA network (which is what they found). This very important point is not even mentioned until Lines 999-1010 in the Discussion. I recommend that you mention this much earlier in the paper, probably in the Abstract and Introduction.

2. You place a substantial amount of faith in the “explanatory” powers of your BRTs. This needs to be tempered a fraction. Table 4 indicates that the deviance explained by the BRTs is 29.47% (Regional), 35.73% (PV), 32.66% (PA) and 15.74% (BM). If I understand Figures 7-10 and Table 4 correctly, this amount of deviance explained is then partitioned among 14 potential explanatory predictor variables. Thus, Distance from Port, your strongest driver in the Regional analysis, explains 22.4% of 29.47% (i.e. 6.6%) of the variance. For BM, your strongest driver, Longitude, explains about 17% of 15.74% (i.e. 2.7%). Clearly all of the weaker drivers “explain” very small percentages of the spatial trends. Thus, describing small peaks and troughs in the trends shown in individual panels in Figs. 7-10 is almost describing details unnecessarily. That said, I agree that the major 6 drivers in the BRTs are as you indicate at Lines 1045-47.

3. The spatial scale at which you measure effort (2.25 km2) may be rather coarse to be making confident statements about the lack of evidence for “fishing the line”. Many of the studies of spillover (see references cited below at Line 86) often report this effect at much smaller spatial scales than this. You should at least acknowledge this point.

4. The Discussion is far too long and repetitive (17 pages, with a Summary of almost 7 pages). This should be condensed considerably.

5. Lines 93-97 (Introduction) and 952-954 (Discussion) “…to our knowledge, no study has examined yet how fishers respond to those situations in which they have to cope simultaneously with implementation of an MPA, and with the interaction of external drivers…”. A relevant, similar, example is the perceived effect of the rezoning of Australia’s Great Barrier Reef Marine Park in 2004 on local fisheries described by Fletcher WJ et al (2015) Large-scale expansion of no-take closures within the Great Barrier Reef has not enhanced fishery production. Ecol. Appl. 25: 1187-1196 and critiqued by Hughes TP et al (2016) A critique of claims for negative impacts of marine protected areas on fisheries. Ecol. Appl. 26: 637-641. I would recommend that you cite these two papers.

Minor Comments.

Abstract.

Line 30. MP Area (omit s).

Line 31. Note change in font size of text at full stop.

Line 37. Unfeasible (not infeasible).

Introduction

Line 73. “…pay greater attention to the human dimensions of MPAs [10,11]…” In addition references 10 and 11 cited, both by the authors of the current paper, a very relevant example possibly worth citing here would be: Alcala A.C. and G.R. Russ (2006). No-take marine reserves and reef fisheries management in the Philippines: A new people power revolution. Ambio 35(5): 245-254.

Line 86 (and 198). In addition to the Kellner reference (15) on spillover and fishing the line, which is a modelling paper, and Ref. 26 (line 198) and Ref. 70 (line 1264) regarding spillover, three excellent empirical papers on spillover that could be cited are the review by Halpern BS et al (2010) Spillover from marine reserves and the replenishment of fished stocks. Env. Cons. 36: 268-276; Goni R et al (2010) Net contribution of spillover from a marine reserve to fishery catches. Mar. Ecol. Prog. Ser. 400:233-243 (on lobsters in the Mediterranean); and Kerwath SE et al (2013) Marine protected area improves yield without disadvantaging fishers. Nature Communications 4:2347. You should also acknowledge the possibility of larval (as opposed to adult) export from reserves to fished areas, for example: Harrison HB et al (2012) Larval export from marine reserves and the recruitment benefit for fish and fisheries. Current Biology 22:1023-1028.

Lines 93-97. Note major point 5 above.

Lines 116-117. Indicate here the year when the MPAs were implemented (2000).

Materials and Methods.

Line 198. “….and spillover to fishing grounds may occur ([26]”

Lines 208, 209. Tourist or tourism (not touristic).

Lines 273-274. Why calculate effort by dividing catch by catch-per-unit-effort (CPUE)? Surely you measured catch and effort directly to calculate CPUE?

Line 302. ..affected by the potential drivers (add potential).

Lines 310-312. You make it clear that the re-zoning was confounded by the sea-cucumber over-exploitation phase (see also Table 3). Thus, when you talk of changes to effort associated with the zoning (e.g. Lines: 600-605, 989-990, 1035-1036) you must acknowledge this confounding. At lines 1096-1099 you DO acknowledge the confounding, and should in other places in the manuscript.

Lines 367, 401, 407. Insert “the” before: normality assumption, input field, z score.

Results.

Line 505. (Fig. 2a, b, c) should read (Fig. 2d, e, f).

Line 520. (Fig. 2d, e, f) should read (Fig. 2a, b, c).

Line 601. Acknowledge confounding of zoning and sea-cucumber over-exploitation phase.

Line 644. ..the eastern part, ..the southeastern part (insert the).

Line 678. Fishers (add s).

Line 702. These types of fishers..

Line 718. Suggest (not suggests).

Line 888. Western side of

Discussion.

Lines 952-954. Note comment re Fletcher et al (2015) and Hughes et al (2016) above.

Lines1174-1176. Good point. The lobster recovery may not be related to the implementation of the MPAs.

Lines 1188-1190 and 1218-1220. When suggesting a re-evaluation of the MPA zoning, you must be clear about why the MPAs were established: conservation, fisheries management, or both.

Lines 1194. The TURFS suggestion outside the MPAs is a good one.

Lines 1199-1201. Alcala and Russ (2006) could be cited here also.

Lines 1224-1227. Why would an MPA network placed in a biased manner help the fishery if it was set up to avoid the fishery?

Line 1238. ..replicates.

Line 1241. Thirdly (not Fourthly).

Lines 1249-1262. In addition to the Kay example in the Channel Islands, which is a good one, you could also mention the Goni et al (2010) lobster example from the Mediterranean.

Line 1264. Ref. 70 in support of the idea of spillover is inadequate. See references to cite on spillover suggested above.

Line 1291. To support (not the support).

Fig. 2. What do the dark grey and light grey shaded areas of time represent? El Nino/La Nina? Specify in caption.

Figs. 3 and 4. What are the units here? Effort (diver hours)? Specify in caption.

Figs. 3-6. I find it difficult to differentiate Fig. 3 from 4, or Fig. 5 from Fig. 6, simply by eye.

Figures 7-10. Specify acronyms for all of the predictor variables in the caption of Fig. 7, then refer to this in the captions of Figs. 8-10. Reader must be reminded what these variables are in the caption.

Fig. 7. I agree, NearNTZ has no pattern.

Table 1. Caption. Sampling method (not smapling).

Table 3. Caption Line 2: occurring (not occurred).

Table 4. Perhaps call the variables “Predictor Variables” in the caption?

Reviewer #2: General comments:

In this study, the authors aim to investigate the effects of management, biophysical data and socioeconomic factors on the distribution of fishing effort. They use a variety of analytical tools to detect global and local drivers, from the Global Financial Crisis and climatic drivers to the distribution of MPAs. Given the need to better understand drivers of social and ecological dynamics, it will be good to see this paper published. There are two primary concerns that need to be addressed, however.

The first (and most serious) is that there is no mention of overfishing as a possible driver. This may be hard to measure, but in any boom-and-bust dynamic this must be one of the factors investigated. By reading this manuscript, the reader has no idea what kind of fishing effort the spiny lobster and sea cucumber populations in this area can sustainably endure. Ideally, the authors need to weave this consideration into the whole manuscript, and if there is no way of adding actual data on this, they need to make a substantial effort to include information from other studies.

The second is that as it stands, this paper is extremely long and gets way too bogged down in the detail. This whole manuscript needs to be clearly structured and significantly tightened. The introduction neglects to adequately develop the relevant background, and can be much improved with examples and references. The most important points are often lost in the detail, and there is a lot of unnecessary repetition, both between sections and within sections. The authors need to go through the manuscript carefully and re-develop it around the main points they are trying to make.

Further detailed comments are listed below.

Introduction

L55: MPAs more than just a topic of discussion - it would be a stronger opening for your introduction to acknowledge their widespread and increasing implementation.

L65: Change "spatial management and integrated management" to "spatial and integrated management".

You could also briefly mention where Marine Spatial Planning (a term widely used in the Western Pacific) comes in.

L68-69: Please provide one or two examples of this, with references.

L69-72: Please provide one or two examples of this, with references.

L78-79: Who is discussing this? Please provide references. A discussion implies some weighing up of pros and cons; please give examples.

L80: Remove the comma after “grounds”.

L88: Change "on" to "for".

L94: Change the phrase to "...no study has yet examined..."

L96: Remove the "s" from "markets".

L102: Change "on" to "to".

L104: Recommendations cannot be mislead. Perhaps you mean something like "misleading management agencies into making inadequate decisions"?

L110: Remove the comma after “drivers”.

L111: Remove the comma after “MPA”.

L113: The Introduction needs to make a case for why this is a good place for this study. You can use some of the information already in the Methods section, to avoid repetition. I have indicated below with section would fit better here than in the Methods.

L120: The management implications of what / who?

L124: A clearer way to frame the goals of this study, which then can also streamline the structure of the paper, is to pose a list of questions. Then the methods, results and discussion sections can be structured accordingly.

Materials & Methods

L133: Who created this division? Please provide a reference.

L141: Change "in" to "across".

L143: Do you mean that it's officially protected as National Parks, or it's just uninhabited?

L178-232: All this could go in the Introduction. It also needs tightening and streamlining; as it is, it's much too long.

L189: Briefly say what this means for environmental conditions around Galapagos.

L198: Change "spillover" to "spill over". In this context, it's being used as a verb.

L243: This is incorrect - it needs to be expressed as "data points" or "records".

L245: Change "daily-basis" to "daily basis".

L286: This is rather hard to follow. It would be much improved by a table of what data were collected when, and with what method. A little of that is contained within Table 1, but this could be moved to an expanded data collection table.

L316: Does this mean you can't tell which one - stocks collapse or global financial crisis - actually drove the profitability of fishing?

L335: What statistic was used, and what software was used?

L379: These are all examples of goals of your analysis that could be framed as questions and added to the end of the Introduction.

L409: Explain the difference between a hotspot and a cold spot.

L417: Should this be diver hours per unit area?

L452: change "being" to "are".

Results

L524: I don't see these illustrated in the figure. One way to show this in the figure itself would be to add arrows for when these events occurred.

L527: The Results section is not the place to try and find reasons for the results - move all these kinds of inferences to the Discussion section. The Results section is simply for describing results.

L621: These key patterns would be more useful if they were moved to the beginning of each section. The authors could begin with the key patterns and the describe some of the detail.

L625: The best place for this next paragraph would be in the Discussion, where it could then be followed by more detailed discussion about these patterns and their reasons and implications.

L684: The best place for this next paragraph would be in the Discussion, where it could then be followed by more detailed discussion about these patterns and their reasons and implications.

L718-720: This is a very awkward way to start - clearly state your main result. This whole following section is way too long. Please tighten it and clearly highlight the key results that you will discuss in the Discussion sections.

Discussion

L952: To make the reader want to read more, highlight your most important and interesting results at the beginning of the Discussion. You only need one sentence to "sell" the novelty of the methods used.

L976-985: This is what you could start the Discussion with.

L989-997: There's no need to re-iterate detailed results. Stick to discussing them in the context of current knowledge, and the implications of your findings.

L999: Insert "The" at the start of this sentence.

L1017-1029: This seems out of place here. Stick to discussing your results.

L1028: This has already been said. This repetition is not helpful and makes the Discussion hard to read.

L1099: This is a little confusing - the discussion about fishing the line further above suggests that marine zoning was implemented to not affect areas preferred for fishing - but here there's a suggestion that zoning did have a significant effect on fisheries.

L1130: This is good - please develop this further by setting it in context of other studies that may have found similar patterns, and then discuss the implications for management.

Summary, Implications, Conclusions – this back end is far too long, please condense.

Tables and Figures

Table 2: This table really belongs with the paragraphs describing the different factors that have influenced these fisheries.

Figure 2: Throughout the caption, please change "relation" to "relationship". Could you add p-values to the regression figures?

Figure 4 caption: Change "three-time" to "three time".

Figures 8-10: There’s no reason to use the abbreviations in the axis titles. Please write them out in full.

6. PLOS authors have the option to publish the peer review history of their article (what does this mean?). If published, this will include your full peer review and any attached files.

Reviewer #1: No

Reviewer #2: No

---

## [Author Response · Author response to Decision Letter 0]

19 Dec 2019

December 15th, 2019

Heather M. Patterson, Ph.D.

Academic Editor 

PlosOne

Dear Dr. Patterson,

Please find enclosed the revised version of the manuscript entitled “Human and climatic drivers affect spatial fishing patterns in a multiple-use marine protected area: the Galapagos Marine Reserve” by Mauricio Castrejón (corresponding author) and Anthony Charles. 

As required, a revised version of our manuscript that addresses the points raised during the review is submitted as a single Word document file. Given the novel nature of the topic and lack of previous long-term quantitative studies on Galapagos marine zoning, the paper required a longer treatment that the normal article length for PlosOne. Nevertheless, the whole manuscript was edited and reduced, as recommended by reviewers. 

Also please find attached a rebuttal letter that contains our responses to each point raised by the academic editor and two reviewers. We are grateful for the insightful and constructive comments made by all of you. We accepted most of your suggestions. We have written the entire manuscript in a more concise manner, particularly the discussion section. We have also modified some figures and tables as suggested by the referees. We feel that the revision process has greatly enhanced the clarity and quality of the paper. 

The Galapagos National Park Service is the owner of the data used for this paper. Therefore, there are legal restrictions on sharing a de-identified data set. However, the data underlying the results presented in the study are available on request at investigacion@galapagos.gob.ec. We confirm that other researchers would be able to access the data set in the same manner as we did, and we did not have any special access privileges that others would not have.

We hope you will be pleased with this new version of the manuscript. Please do not hesitate to contact me with any questions or concerns. Thank you very much for your consideration. We look forward to receiving the acknowledgment of the manuscript and your decision.

Sincerely yours,

Dr. Mauricio Castrejón

Interdisciplinary PhD Program

Dalhousie University

Halifax, Canada

mauricio.castrejon@dal.ca

PONE-D-19-26580 

Human and climatic drivers affect spatial fishing patterns in a multiple-use marine protected area: the Galapagos Marine Reserve 

Rebuttal letter

We acknowledge useful comments provided by the academic editor and two referees. Most of their suggestions have been included in the revised version of the manuscript. In this letter, we explain how and where the corrections have been incorporated into the text. We follow the editorial order and have numbered referees' suggestions and comments.

Editorial comments

The issue with this manuscript is that it is just far too long and wordy. It would be much more accessible for the reader if it was written in a more concise manner. Instead, it is very repetitive and often includes details that are not germane to the actual study. There are also too many figures and some should be moved to the Supporting Information, too many sub-headings in the methods, results and discussion (every paragraph does not need its own sub-heading) and too many acronyms, that are often not used or are redefined over and over which is unnecessary. Please only introduce an acronym if it is used several times again in the manuscript and then use it consistently; it only has to be defined once. The Discussion is a particular problem and is 17 pages on its own, which is just far too long and there are a number of paragraphs that add little and should be deleted. The text overall needs work and contains quite a few grammatical errors.

R: We have written the entire manuscript in a more concise manner, particularly the discussion section. We have also modified some figures and tables as suggested by the referees. The number of sub-headings in the Methods, Results and Discussion sections was reduced, as well as the number of acronyms. Grammatical errors were corrected.

Line 30: Do not capitalize ‘marine protected areas’

R: Suggestion taken. 

Line 31: Looks like the font size at the end of this line changes

 R: Font size changed.

Line 35: Write as ‘interpretation of assessments of’

R: Suggestion taken. 

Line 36: delete ‘assessments’

R: Suggestion taken. 

Line 39: ‘long-term, spatially-explicit’

R: Suggestion taken. 

Line 43: delete ‘(GIS)’ here as this is not used again in the abstract

R: Suggestion taken. 

Line 55: Do not capitalise ‘protected areas’

R: Suggestion taken. 

Line 64: ‘spatially-demarcated areas’

R: Suggestion taken. 

Lines 70-71: don’t understand this text offset by en dashes. Doesn’t follow the previous text so either delete or rewrite.

R: Text was edited. 

Line 72: Should the ‘and’ before ‘policy’ be deleted? Doesn’t make sense as written.

R: Text was edited. 

Line 73: replace the hyphen with a comma and delete the comma after ‘countries’

R: Suggestion taken. 

Line 80: delete the comma

R: Suggestion taken. 

Line 94: no study has yet examined’

R: Suggestion taken. 

Line 95: delete the comma after ‘MPA’

R: Suggestion taken. 

Lines 104-105: Not sure what this means ‘potentially misleading….’ How can you mislead something that has been adopted? Please clarify or delete.

R: Text was edited. 

Line 110: Delete the comma

R: Suggestion taken. 

Line 11: Delete the comma

R: Suggestion taken. 

Lines 112-113: Move this to the beginning of the sentence ‘The Galapagos Marine Reserve…..’

R: Suggestion taken. 

Line 135: Add ‘the’ before ‘El Nino’

R: Suggestion taken. 

Lines 137: Move the ‘according to Edgar’ out of the main title of the figure. Either put it in the notes or just include it in the text.

R: Suggestion taken. Cite included in the text. 

Line 141: Write as 25,144 and ‘distributed on’

R: Suggestion taken. 

Line 150: Delete ‘to’ after ‘provides’

R: Suggestion taken. 

Line 153: Write as ‘large, wooden boats’

R: Suggestion taken. 

Line 169: Write as ‘fishing season since 1999’

R: Suggestion taken. 

Line 170: delete ‘since 1999’

R: Suggestion taken. 

Lines 178-232: This is very long and wordy and much of it does not seem directly relevant to the study. I would cut this back to only the information that is applicable to the study.

R: Suggestion taken. Paragraph was reduced and edited. 

Line 179: The abbreviation GMR has already been established and does not need to be established here again so just use ‘GMR’

R: Suggestion taken. 

Line 184: The colon should be a comma

R: Text was edited. 

Line 184: Do not introduce the abbreviation PMB

R: Text was edited.

Line 185: Do not introduce the abbreviation IMA

 R: Text was edited.

Line 189: 1997-98

R: Text was edited.

Line 201: Delete the apostrophe in miles

R: Text was edited.

Line 208: Should be ‘tourist activities’ and would be good to have an example of what that means.

R: Text was edited.

Line 209: tourist activities

R: Text was edited.

Line 216: Need to define CPUE here and define the abbreviation if you are going to use it

R: Text was edited.

Lines 224-232: Most of this is a repeat from the introduction on why the study is being done so delete

R: This paragraph was deleted.

Line 238: ‘spatially-explicit’

R: Suggestion taken. 

Line 242: ‘Fisheries-related data’

R: Suggestion taken. 

Line 245: Table 1

R: Suggestion taken. 

Line 258: delete ‘up’

R: Suggestion taken. 

Line 263: 17,723

R: Suggestion taken. 

Line 272: ‘from 2009’

R: Suggestion taken. 

Line 274: Need to define CPUE at line 216, not here, so just use ‘CPUE’

R: Text edited. 

Line 283: delete the comma at the end of the line

R: Suggestion taken. 

Line 290: Too many sub-headings, should delete some

R: Suggestion taken. The number of sub-headings was reduced. 

Line 292: I would say ‘The most potentially relevant’ because there is no evidence these are relevant.

R: Suggestion taken. Text edited. 

Line 296: Again ‘that potentially affected’

R: Suggestion taken. Text edited.

Line 307: ‘boom-and-bust’

R: Suggestion taken. 

Line 312: delete ‘the Galapagos’ as this is not necessary

R: Suggestion taken. 

Line 313: delete the comma

R: Suggestion taken. 

Line 317: Again, delete ‘in Galapagos’

R: Suggestion taken. 

Line 343: Don’t need a part 1 and 2 for the sub-heading

R: Suggestion taken. Sub-headings were edited. 

Line 346: abbreviation GIS has already been established and does not need to be established here again so just use ‘GIS’

R: Suggestion taken. 

Line 357: Delete the apostrophe

R: Suggestion taken. 

Lines 370-371: Don’t need a new sub-heading

R: Suggestion taken. Sub-heading deleted.

Line 374: Should be ‘is concentrated’

R: Suggestion taken. 

Line 388: ‘fine-scale distribution’

R: Suggestion taken. 

Line 407: Why are there vertical lines around 1.96?

R: Vertical lines deleted. 

Lines 447-457: Is all this background on BRT required?

R: As BRT model are relatively a new analytical tool, BRT background is needed. Nevertheless, text was reduced and edited. 

Lines 450-451: Do not capitalise ‘general linear models’ or ‘general additive models’ and if the abbreviations are not used again delete them

R: Suggestion taken. 

Line 489: results should be in the past tense so please check the text

R: Suggestion taken. Text was revised and edited.

Line 507: Long-term variation

R: Suggestion taken. 

Line 524: This is confusing as it is not clear to me how the authors have decided that the GFC contributed to this. Is this just due to the time overlap? Not sure how Fig 2 demonstrates this either.

R: Figure 2 and text were edited to explain how fishing capacity varied during the five periods analyzed, including GFC. 

Line 531: Use the abbreviation GMR

R: Suggestion taken. 

Line 534: Don’t need a part 1 and 2 

R: Suggestion taken. 

Line 561: The authors introduced an abbreviation for core area and distribution ranges on line 539 so they should use them throughout the manuscript or do not introduce the abbreviation

R: Suggestion taken. Abbreviations were eliminated. 

Line 564: Should be ‘were shared’

R: Suggestion taken. 

Line 566: Should be ‘showed’

R: Suggestion taken. 

Line 571: Should be ‘trends of the’

R: Suggestion taken. 

Lines 588-589: This is very awkward and the results section is not the place for this. Just write is as a simple statement of the results (not in italics) ‘Based on these results there is a pattern where…..’

R: Text was edited. 

Lines 619-620: Same comment as above.

R: Text was edited. 

Line 626: ‘boom-and-bust’

R: Suggestion taken. 

Line 630: Replace ‘In consequence’ with ‘Therefore’

R: Suggestion taken. 

Line 634: Don’t need a part 2

R: Suggestion taken. 

Line 686: ‘fishing effort was concentrated’ and delete the apostrophe at the end of the line.

R: Suggestion taken. 

Line 688: boom-and-bust

R: Suggestion taken. 

Line 698: replace the dash with a comma

R: Suggestion taken. 

Line 702: ‘This type of fisher’

R: Suggestion taken. 

Line 721: pearson’s

R: Suggestion taken. 

Line 724: 35.73%, 32.66%

R: Suggestion taken. 

Line 725: Pearson’s

R: Suggestion taken. 

Line 728: delete ‘VI’

R: Suggestion taken. 

Line 731: Use the abbreviation BRT

R: Suggestion taken. 

Line 745: Just use the abbreviation RN

R: Suggestion taken. 

Line 768: Need a comma after ‘islands’

R: Suggestion taken. 

Lines 781-782: This is confusing as written. Notes there was a decline but the revenue goes up to $11,000? Also, write as US$2000 etc. Please change throughout manuscript.

R: Text was edited to improve understanding. 

Line 793: delete ‘VI’

R: Suggestion taken. 

Line 811: replace the dash with a comma

R: Suggestion taken. 

Line 819: delete ‘the random number variable’

R: Suggestion taken. 

Line 829: Replace the dashes with commas

R: Suggestion taken. 

Line 845: Delete ‘a value of’

R: Suggestion taken. 

Line 854: delete ‘VI’

R: Suggestion taken. 

Line 860: delete ‘VI’

R: Suggestion taken. 

Line 866: delete ‘VI’

R: Suggestion taken. 

Line 870: Write as US$2500

R: Suggestion taken. 

Line 871: same as above

R: Suggestion taken. 

Line 873: same as above

R: Suggestion taken. 

Line 886: Should be ‘islands’

R: Suggestion taken. 

Line 942: delete the apostrophe and write as ‘fishing fleet core areas’

R: Suggestion taken. 

Line 944: ‘boom-and-bust’

R: Suggestion taken. 

Line 950: Too many sub-headings

R: Suggestion taken. Sub-headings were reduced. 

Lines 956-962: This text is repetitive and unnecessary so delete and move the next paragraph up to follow the opening sentence of the paragraph.

R: Suggestion taken. Text was deleted and Discussion was edited. 

Lie 970: ‘boom-and-bust’

R: Suggestion taken. 

Line 981: GMR

R: Suggestion taken. 

Line 985: ‘fishing fleet core areas’

R: Suggestion taken. 

Line 994: Don’t redefine the abbreviation nearNTZ, just use it.

R: Suggestion taken. 

Line 1000: GMR

R: Suggestion taken. 

Line 1008: Move ‘historically’ to line 1009 after ‘areas’

R: Suggestion taken. 

Line 1045: ‘these adaptive responses’? what responses? 

R: Text edited. 

Line 1048: ‘although this was not the case for Puerto Ayora’

R: Suggestion taken. 

Line 1053: delete ‘on the water’

R: Suggestion taken. 

Line 1065: Reference not formatted correctly, needs a reference number

R: References were formatted. 

Line 1066: Do not capitalise ‘discrete choice models’ and delete the acronym as it is not used again.

R: Suggestion taken. 

Line 1074: Replace the dash with a comma

R: Suggestion taken. 

Line 1075: replace the dash with a comma

R: Suggestion taken. 

Line 1082: Delete ‘fishing fleets’

R: Suggestion taken. 

Line 1086: Reference not formatted correctly, needs a reference number

R: References were formatted. 

Line 1111: delete the comma and ‘on the water’

R: Suggestion taken. 

Line 1133: CPUE has already been defined and does not need to be defined again here so just use CPUE

R: Suggestion taken. 

Line 1145: Delete ‘has’

R: Suggestion taken. 

Line 1149: delete ‘probably’

R: Suggestion taken. 

Line 1150: add ‘likely’ before ‘irrelevant’

R: Suggestion taken. 

Lines 1153-1165: This is very confusing. I have no idea why the authors are providing a summary half way through the Discussion. This is unnecessary and repetitive so please delete.

R: Suggestion taken. This paragraph was deleted. 

Line 1166: Use ‘GMR’

R: Suggestion taken. 

Line 1173: Use ‘GMR’

R: Suggestion taken. 

Lines 1178-1183: This is a sentence, not a paragraph, so please move it up to join the paragraph above.

R: Suggestion taken. 

Line 1178: Delete both commas

R: Suggestion taken. 

Line 1180: Use ‘GMR’

R: Suggestion taken. 

Line 1185: replace the dash with a comma

R: Suggestion taken. 

Line 1187: replace the dash with a comma

R: Suggestion taken. 

Line 1189: Use ‘GMR’

R: Suggestion taken. 

Line 1192: Use ‘GMR’

R: Suggestion taken. 

Line 1197: Use ‘GMR’

R: Suggestion taken. 

Line 1205: Do not introduce the abbreviation MSC

R: Suggestion taken. 

Lines 1208-1216: Move this up to join the previous paragraph (which is actually a sentence).

R: Suggestion taken. 

Line 1210: Use ‘GMR’

R: Suggestion taken. 

Line 1222: Replace ‘usefulness’ with ‘utility’

R: Suggestion taken. 

Line 1223: EBM has already been defined and does not need to be defined again here so just use EBM

R: Text was edited. 

Line 1226: Use ‘GMR’

R: Suggestion taken. 

Line 1231: replace ‘over’ with ‘on’ and do not need to provide the scientific name as this has already been established

R: Suggestion taken. 

Line 1235: Delete ‘Using non-parametric statistics’

R: Suggestion taken. 

Line 1253: Do not introduce the abbreviation SBCI

R: Suggestion taken. 

Line 1254: replace ‘over’ with ‘on’ and abbreviate as P. interruptus

R: Suggestion taken. 

Line 1257: Should be vs.

R: Suggestion taken. 

Line 1259: Don’t need z=0.59

R: Text was edited.

Line 1260: Spell out SBCI

R: Text was edited

Lines 1263-1270: This is very repetitive and adds nothing to the discussion in my view so I would delete.

R: Suggestion taken. These lines were deleted.

Line 1283: Delete ‘enabling’

R: Suggestion taken. 

Line 1290: Delete ‘enabling’

R: Suggestion taken. 

Line 1293: ‘based on the core areas and distribution ranges of fishing fleets’

R: Suggestion taken. 

Line 1303: Delete ‘fishing fleets’

R: Suggestion taken. 

Lines 1305-1315: Again, I find this text unnecessary and it could be deleted.

R: Text was deleted. 

Lines 1319-1323: This is a sentence, not a paragraph

R: Text was edited. 

Line 1321: EBM has already been defined and does not need to be defined again here so just use EBM

R: Text was edited. 

Line 1328: ‘should not be taken for granted; MPAs are not a panacea. (delete the rest ‘i.e. a one-size…..’

R: Suggestion taken. 

Line 1332: ‘with global climate change’

R: Suggestion taken. 

Line 1336: delete ‘learned’ and write as ‘is that assessments of MPA effectiveness’

R: Suggestion taken. 

Reviewer 1

This study evaluated how the spatiotemporal allocation of fishing effort for lobsters in the Galapagos multiple-use Marine Protected Area was affected by the interaction of diverse climatic and human drivers, before and after implementation of no-take zones. The study used GIS data on fishing effort and BRTs to attempt to identify how these drivers affected spatial fishing patterns. The paper concludes that the boom-and-bust exploitation of the sea cucumber fishery and the global financial crisis (2007-2009), rather than no-take zone implementation, were the most important drivers affecting the distribution of fishing effort for lobsters across the archipelago. The study is spatially and temporally extensive (most of the Galapagos Islands, 1997-2011), the data are fairly well-analyzed and interpreted, and the manuscript is well-written. I have no major disagreements with the conclusions. I also have some sympathy with the suggestions that the MPA network placement could be revisited or even revised, and outside the network TURFS encouraged. My comments are mostly to assist the authors with publication.

Major Comments.

1. This is a paper that, in effect, quantifies spatial and temporal trends in fishing effort of a lobster fishery in a developing country. Yet the emphasis chosen is how this data informs effects of MPA implementation. It is highly commendable that the study includes before and after implementation data. In fact, this is such an important aspect of the study, I would recommend that the authors stress this point more in the paper. However, this MPA network is also well-known as a “classic” case where fishers ensured that no-take zones were NOT placed where fishers fished (Edgar et al 2004 Ref. 22 in this manuscript). That is, it is a case where you might NOT expect much change in spatial effort in the lobster fishery pre- and post-implementation of the MPA network (which is what they found). This very important point is not even mentioned until Lines 999-1010 in the Discussion. I recommend that you mention this much earlier in the paper, probably in the Abstract and Introduction.

R: Suggestion taken. Both points suggested are mentioned in the Introduction. 

2. You place a substantial amount of faith in the “explanatory” powers of your BRTs. This needs to be tempered a fraction. Table 4 indicates that the deviance explained by the BRTs is 29.47% (Regional), 35.73% (PV), 32.66% (PA) and 15.74% (BM). If I understand Figures 7-10 and Table 4 correctly, this amount of deviance explained is then partitioned among 14 potential explanatory predictor variables. Thus, Distance from Port, your strongest driver in the Regional analysis, explains 22.4% of 29.47% (i.e. 6.6%) of the variance. For BM, your strongest driver, Longitude, explains about 17% of 15.74% (i.e. 2.7%). Clearly all of the weaker drivers “explain” very small percentages of the spatial trends. Thus, describing small peaks and troughs in the trends shown in individual panels in Figs. 7-10 is almost describing details unnecessarily. That said, I agree that the major 6 drivers in the BRTs are as you indicate at Lines 1045-47.

R: The deviance explained is not partitioned among the explanatory predictor variables. Table 4 and Figures 7-10 show the variable importance (VI), which was estimated by averaging the number of times a variable is selected for splitting and the squared improvement resulting from these splits. VI scores provide a measure of the relative influence of predictor variables used to build the model. Values are scaled so that the sum adds to 100, with higher numbers indicating a stronger influence on the response variable. This explanation is provided in Lines 730-735.

3. The spatial scale at which you measure effort (2.25 km2) may be rather coarse to be making confident statements about the lack of evidence for “fishing the line”. Many of the studies of spillover (see references cited below at Line 86) often report this effect at much smaller spatial scales than this. You should at least acknowledge this point.

R: We agree that a finer scale probably would be needed to evaluate a spiny lobster spillover effect, but this is not the objective of this study. We evaluated finer and coarser spatial scales to conduct the hotspot analysis and the 2.25km2 scale was the most proper scale to visualize the results and to evaluate the presence of a fishing the line effect around no-take zones. On the other hand, a fishing the line effect around the Galapagos Marine Reserve was detected by Bucaram at al. 2018 using a coarser scale of analysis (see “Assessing fishing effects inside and outside an MPA: The impact of the Galapagos Marine Reserve on the Industrial pelagic tuna fisheries during the first decade of operation”). Therefore, we think that a 2.25km2 is an appropriate spatial scale of analysis for our case study. 

4. The Discussion is far too long and repetitive (17 pages, with a Summary of almost 7 pages). This should be condensed considerably.

 R: Suggestion taken. Discussion was considerably reduced and edited. 

5. Lines 93-97 (Introduction) and 952-954 (Discussion) “…to our knowledge, no study has examined yet how fishers respond to those situations in which they have to cope simultaneously with implementation of an MPA, and with the interaction of external drivers…”. A relevant, similar, example is the perceived effect of the rezoning of Australia’s Great Barrier Reef Marine Park in 2004 on local fisheries described by Fletcher WJ et al (2015) Large-scale expansion of no-take closures within the Great Barrier Reef has not enhanced fishery production. Ecol. Appl. 25: 1187-1196 and critiqued by Hughes TP et al (2016) A critique of claims for negative impacts of marine protected areas on fisheries. Ecol. Appl. 26: 637-641. I would recommend that you cite these two papers.

R: We reviewed Fletcher et al (2015) and, even though they recognize that MPAs can be affected by diverse drivers of change, they do not evaluate their impact in a quantitative way as we did. Therefore, we edited the text in the following way: “…to our knowledge, no study has examined yet, in a quantitative way, how fishers respond to those situations in which they have to cope simultaneously with implementation of an MPA, and with the interaction of external drivers…”.

Minor Comments. Abstract.

Line 30. MP Area (omit s).

 R: Suggestion rejected. MPAs is a term commonly used in the scientific literature. 

Line 31. Note change in font size of text at full stop. 

R: Suggestion taken. 

Line 37. Unfeasible (not infeasible).

R: Suggestion taken. 

Introduction

Line 73. “…pay greater attention to the human dimensions of MPAs [10,11]…” In addition references 10 and 11 cited, both by the authors of the current paper, a very relevant example possibly worth citing here would be: Alcala A.C. and G.R. Russ (2006). No-take marine reserves and reef fisheries management in the Philippines: A new people power revolution. Ambio 35(5): 245-254.

R: Suggestion taken. Cite added. 

Line 86 (and 198). In addition to the Kellner reference (15) on spillover and fishing the line, which is a modelling paper, and Ref. 26 (line 198) and Ref. 70 (line 1264) regarding spillover, three excellent empirical papers on spillover that could be cited are the review by Halpern BS et al (2010) Spillover from marine reserves and the replenishment of fished stocks. Env. Cons. 36: 268-276; Goni R et al (2010) Net contribution of spillover from a marine reserve to fishery catches. Mar. Ecol. Prog. Ser. 400:233-243 (on lobsters in the Mediterranean); and Kerwath SE et al (2013) Marine protected area improves yield without disadvantaging fishers. Nature Communications 4:2347. You should also acknowledge the possibility of larval (as opposed to adult) export from reserves to fished areas, for example: Harrison HB et al (2012) Larval export from marine reserves and the recruitment benefit for fish and fisheries. Current Biology 22:1023-1028.

R: Suggestion taken. We replace Kellner reference by Halpern et al (2010) and Kerwarth et al (2013). Line 198 was eliminated. 

Lines 93-97. Note major point 5 above.

 R: Same response as in point 5 above.

Lines 116-117. Indicate here the year when the MPAs were implemented (2000).

R: Suggestion taken. Text was edited.

Materials and Methods.

Line 198. “….and spillover to fishing grounds may occur ([26]” Lines 208, 209. Tourist or tourism (not touristic).

R: Suggestion taken.

Lines 273-274. Why calculate effort by dividing catch by catch-per-unit-effort (CPUE)? Surely you measured catch and effort directly to calculate CPUE?

R: We estimated total fishing effort by dividing total catch by average CPUE per fishing season. We conducted this analysis to estimate the amount of sampling effort per fishing season. For all analysis presented in the paper, we used CPUE data estimated directly from catch and effort data per fishing trip. 

Line 302. ..affected by the potential drivers (add potential).

R: Suggestion taken.

Lines 310-312. You make it clear that the re-zoning was confounded by the sea- cucumber over-exploitation phase (see also Table 3). Thus, when you talk of changes to effort associated with the zoning (e.g. Lines: 600-605, 989-990, 1035- 1036) you must acknowledge this confounding. At lines 1096-1099 you DO acknowledge the confounding, and should in other places in the manuscript.

R: Suggestion taken. Text was edited.

Lines 367, 401, 407. Insert “the” before: normality assumption, input field, z score.

R: Suggestion taken.

Results.

Line 505. (Fig. 2a, b, c) should read (Fig. 2d, e, f). Line 520. (Fig. 2d, e, f) should read (Fig. 2a, b, c).

R: Suggestion taken.

Line 601. Acknowledge confounding of zoning and sea-cucumber over-exploitation phase.

R: Suggestion taken.

Line 644. ..the eastern part, ..the southeastern part (insert the). Line 678. Fishers (add s).

R: Suggestions taken.

Line 702. These types of fishers. Line 718. Suggest (not suggests). Line 888. Western side of 

R: Suggestion taken.

Discussion.

Lines 952-954. Note comment re Fletcher et al (2015) and Hughes et al (2016) above.

 R: Same response as in point 5 above.

Lines1174-1176. Good point. The lobster recovery may not be related to the implementation of the MPAs.

 R: No response needed. 

Lines 1188-1190 and 1218-1220. When suggesting a re-evaluation of the MPA zoning, you must be clear about why the MPAs were established: conservation, fisheries management, or both.

R: Suggestions taken. Text was edited as: “…we suggest re-evaluating the distribution of no-take zones across the GMR to promote the sustainability of the spiny lobster fishery and conserve key biodiversity areas”.

Lines 1194. The TURFS suggestion outside the MPAs is a good one. Lines 1199-1201. Alcala and Russ (2006) could be cited here also.

R: Suggestions taken. Cite added. 

Lines 1224-1227. Why would an MPA network placed in a biased manner help the fishery if it was set up to avoid the fishery?

R: We agree. However, there was no scientific evidence about the long-term impact of no-take zones on the fishing effort dynamic for the spiny lobster fishery before this study. Our study highlights the need to redistribute no-take zones to accomplish conservation and fishery management objectives. 

Line 1238. ..replicates.

R: Suggestion taken. 

Line 1241. Thirdly (not Fourthly).

R: Suggestion taken.

Lines 1249-1262. In addition to the Kay example in the Channel Islands, which is a good one, you could also mention the Goni et al (2010) lobster example from the Mediterranean.

R: During the edition of the Discussion, we decided to eliminate Lines 1249-1262. We took this decision to put more emphasis in the discussion of other results directly associated to the objectives of our study. 

Line 1264. Ref. 70 in support of the idea of spillover is inadequate. See references to cite on spillover suggested above.

R: The academic editor suggested to eliminate Lines 1263-1270. We accepted this suggestion as we decide to put more emphasis in other results of the paper. 

Line 1291. To support (not the support).

R: Suggestion taken.

Fig. 2. What do the dark grey and light grey shaded areas of time represent? El Nino/La Nina? Specify in caption.

R: Dark grey and light grey areas represents the periods analyzed. Figure 2 has been edited. The name of the periods was added. Caption was edited. 

Figs. 3 and 4. What are the units here? Effort (diver hours)? Specify in caption.

R: Standard deviation ellipses (SDE) polygons represent graphical summaries of the central tendency, dispersion and directional trends of fishing fleets. Core areas and distribution ranges refer to those areas covering 68% (1 SDE) and 95% (2 SDE) of the full spatial extent of fishing fleet distribution, respectively. Explanation is provided in Lines 324-327 of the revised manuscript. 

Figs. 3-6. I find it difficult to differentiate Fig. 3 from 4, or Fig. 5 from Fig. 6, simply by eye.

R: Text was edited to improve explanation of spatial patterns. References to Figures 3-6 was added to the text. 

Figures 7-10. Specify acronyms for all of the predictor variables in the caption of Fig. 7, then refer to this in the captions of Figs. 8-10. Reader must be reminded what these variables are in the caption.

 R: Full name of acronyms was added to caption of Figures 7-10.

Fig. 7. I agree, NearNTZ has no pattern.

 R: No response needed. 

Table 1. Caption. Sampling method (not smapling). Table 3. Caption Line 2: occurring (not occurred).

R: Text edited.

Table 4. Perhaps call the variables “Predictor Variables” in the caption?

R: Suggestion taken.

Reviewer 2

General comments

In this study, the authors aim to investigate the effects of management, biophysical data and socioeconomic factors on the distribution of fishing effort. They use a variety of analytical tools to detect global and local drivers, from the Global Financial Crisis and climatic drivers to the distribution of MPAs. Given the need to better understand drivers of social and ecological dynamics, it will be good to see this paper published. There are two primary concerns that need to be addressed, however.

The first (and most serious) is that there is no mention of overfishing as a possible driver. This may be hard to measure, but in any boom-and-bust dynamic this must be one of the factors investigated. By reading this manuscript, the reader has no idea what kind of fishing effort the spiny lobster and sea cucumber populations in this area can sustainably endure. Ideally, the authors need to weave this consideration into the whole manuscript, and if there is no way of adding actual data on this, they need to make a substantial effort to include information from other studies.

R: As the objective of this study was to predict fishing effort distribution rather than catch or catch-per-unit-effort (CPUE), we focused our analysis on the human element (effort), rather than the interaction between humans and the target species themselves (catch or CPUE). This approach helped us to simplify the interpretation of the results and more accurately predict fishing effort. On the other hand, as overfishing of the spiny lobster fishery is the consequence of the external drivers of change analyzed in this study, particularly of the boom-and-bust exploitation of the sea cucumber fishery, we did not consider overfishing explicitly as a driver of change. Nevertheless, we analyzed the factors influencing fishing effort, including the previous lobster catch and sea cucumber revenues, which were relevant as fishing effort predictors. These two predictors were affected by the overexploitation of the sea cucumber and spiny lobster fisheries. We edited the text to highlight this fact in the manuscript (Lines 1988-1991 of the revised manuscript). In addition, we explained in the Discussion that the spiny lobster fishery was overexploited due to the overcapitalization caused by the expansion of the sea cucumber fishery and explained the consequences of overexploitation on fishing capacity (Lines 1858-1853).

The second is that as it stands, this paper is extremely long and gets way too bogged down in the detail. This whole manuscript needs to be clearly structured and significantly tightened. The introduction neglects to adequately develop the relevant background, and can be much improved with examples and references. The most important points are often lost in the detail, and there is a lot of unnecessary repetition, both between sections and within sections. The authors need to go through the manuscript carefully and re-develop it around the main points they are trying to make.

R: We have written the entire manuscript in a more concise manner, particularly the discussion section. Introduction was improved by including relevant background about the Galapagos Marine Reserve. The Methods, Results and Discussion sections were restructured, reduced and edited to avoid unnecessary repetition. 

Further detailed comments are listed below.

Introduction

L55: MPAs more than just a topic of discussion - it would be a stronger opening for your introduction to acknowledge their widespread and increasing implementation.

R: Suggestion taken. Introduction was edited following reviewer’s suggestions. 

L65: Change "spatial management and integrated management" to "spatial and integrated management".You could also briefly mention where Marine Spatial Planning (a term widely used in the Western Pacific) comes in.

R: Text edited. We did not mention the term Marine Spatial Planning. Instead, we edited the introduction to put more emphasis on the widespread and increasing implementation of MPAs and to explain in a better way the justification for this study. 

L68-69: Please provide one or two examples of this, with references.

R: These lines were eliminated, but additional references were added at the beginning of the introduction. 

L69-72: Please provide one or two examples of this, with references.

R: These lines were eliminated, but additional references were added at the beginning of the introduction. 

L78-79: Who is discussing this? Please provide references. A discussion implies some weighing up of pros and cons; please give examples.

R: These lines were eliminated and Introduction was edited. 

L80: Remove the comma after “grounds”.

R: Suggestion taken.

L88: Change "on" to "for".

R: Suggestion taken.

L94: Change the phrase to "...no study has yet examined..."

R: Suggestion taken.

L96: Remove the "s" from "markets".

R: Suggestion taken.

L102: Change "on" to "to".

R: Suggestion taken.

L104: Recommendations cannot be mislead. Perhaps you mean something like "misleading management agencies into making inadequate decisions"?

R: Text was edited. 

L110: Remove the comma after “drivers”.

R: Suggestion taken.

L111: Remove the comma after “MPA”.

R: Suggestion taken.

L113: The Introduction needs to make a case for why this is a good place for this study. You can use some of the information already in the Methods section, to avoid repetition. I have indicated below with section would fit better here than in the Methods.

R: Suggestion taken. We moved some information from the Method section to the Introduction section to make stronger our case to conduct this study in the Galapagos Marine Reserve. 

L120: The management implications of what / who?

R: Suggestion taken. Management implications for fisheries management. 

L124: A clearer way to frame the goals of this study, which then can also streamline the structure of the paper, is to pose a list of questions. Then the methods, results and discussion sections can be structured accordingly.

R: Suggestion taken. We framed in a better way the goals of this study and restructured the Results and Discussion sections. 

Materials & Methods

L133: Who created this division? Please provide a reference.

R: Suggestion taken. A reference was provided.

L141: Change "in" to "across".

R: Suggestion taken.

L143: Do you mean that it's officially protected as National Parks, or it's just uninhabited?

R: Text edited. It is officially protected as a National Park. 

L178-232: All this could go in the Introduction. It also needs tightening and streamlining; as it is, it's much too long.

R: Suggestion taken. The Method section was reduced. 

L189: Briefly say what this means for environmental conditions around Galapagos.

R: We think that this is not necessary as we mentioned at the beginning of the Method section that El Niño has a strong influence in the abundance and distribution of fish and macro-invertebrate assemblages. 

L198: Change "spillover" to "spill over". In this context, it's being used as a verb.

R: Suggestion taken.

L243: This is incorrect - it needs to be expressed as "data points" or "records".

R: Suggestion taken.

L245: Change "daily-basis" to "daily basis".

R: Suggestion taken.

L286: This is rather hard to follow. It would be much improved by a table of what data were collected when, and with what method. A little of that is contained within Table 1, but this could be moved to an expanded data collection table.

R: Text was reduced and edited for better understanding. In addition, Table S1 provides a summary of the fishery monitoring data gathered for the spiny lobster fishery at the three main ports of the Galapagos Marine Reserve from 1997 to 2011.

L316: Does this mean you can't tell which one - stocks collapse or global financial crisis - actually drove the profitability of fishing?

R: Both drivers caused economic perturbations that affected the profitability of the small-scale fishing sector in Galapagos. For this reason they were grouped together in the same period. 

L335: What statistic was used, and what software was used?

R: Suggestion taken. This information is now provided in the text. 

L379: These are all examples of goals of your analysis that could be framed as questions and added to the end of the Introduction.

R: Suggestion taken. We framed in a better way the goals of this study in the Introduction and restructure the Results and Discussion sections. 

L409: Explain the difference between a hotspot and a cold spot.

R: An explanation is provided in Lines 623-627 of the revised paper. 

L417: Should this be diver hours per unit area?

R: No, just diver-hours. 

L452: change "being" to "are".

R: Suggestion taken.

Results

L524: I don't see these illustrated in the figure. One way to show this in the figure itself would be to add arrows for when these events occurred.

R: Suggestion taken. Figure 2 has been edited. The name of the periods was added. 

L527: The Results section is not the place to try and find reasons for the results - move all these kinds of inferences to the Discussion section. The Results section is simply for describing results.

R: Suggestion taken. Inferences were moved to the Discussion section. 

L621: These key patterns would be more useful if they were moved to the beginning of each section. The authors could begin with the key patterns and the describe some of the detail.

R: Suggestion taken. Results section was edited following reviewer’s comments. 

L625: The best place for this next paragraph would be in the Discussion, where it could then be followed by more detailed discussion about these patterns and their reasons and implications.

R: Suggestion taken. Paragraph eliminated and integrated in the Discussion. 

L684: The best place for this next paragraph would be in the Discussion, where it could then be followed by more detailed discussion about these patterns and their reasons and implications.

R: Suggestion taken. Paragraph eliminated and integrated in the Discussion. 

L718-720: This is a very awkward way to start - clearly state your main result. This whole following section is way too long. Please tighten it and clearly highlight the key results that you will discuss in the Discussion sections.

R: Suggestion taken. Paragraph was reduced and edited. 

Discussion

L952: To make the reader want to read more, highlight your most important and interesting results at the beginning of the Discussion. You only need one sentence to "sell" the novelty of the methods used.

R: Suggestion taken. Discussion was edited to highlight our most important and interesting results.

L976-985: This is what you could start the Discussion with.

R: Suggestion taken. Discussion was edited. 

L989-997: There's no need to re-iterate detailed results. Stick to discussing them in the context of current knowledge, and the implications of your findings.

R: Suggestion taken. Discussion was edited. 

L999: Insert "The" at the start of this sentence.

R: Suggestion taken. 

L1017-1029: This seems out of place here. Stick to discussing your results.

R: Suggestion taken. These lines were edited. 

L1028: This has already been said. This repetition is not helpful and makes the Discussion hard to read.

R: Suggestion taken. Text was edited. 

L1099: This is a little confusing - the discussion about fishing the line further above suggests that marine zoning was implemented to not affect areas preferred for fishing - but here there's a suggestion that zoning did have a significant effect on fisheries.

R: Suggestion taken. Text was edited to improve understanding.

L1130: This is good - please develop this further by setting it in context of other studies that may have found similar patterns, and then discuss the implications for management.

R: Suggestion taken. We have added a paragraph (Lines 910-930) that address the points raised by the reviewer. 

Summary, Implications, Conclusions – this back end is far too long, please condense.

R: Suggestion taken. Text was reduced and edited. 

Tables and Figures

Table 2: This table really belongs with the paragraphs describing the different factors that have influenced these fisheries.

R: Suggestion taken. Table 2 was moved to Study area section. 

Figure 2: Throughout the caption, please change "relation" to "relationship". Could you add p-values to the regression figures?

R: Suggestion taken. P-values were added to regression figures. 

Figure 4 caption: Change "three-time" to "three time".

R: Suggestion taken. 

Figures 8-10: There’s no reason to use the abbreviations in the axis titles. Please write them out in full.

 R: Full names for some predictor variables, such as NearNTZ, are too long to be added to the axis titles. Instead, full name of abbreviations was added to legends of Figures 7-10.

---

## [Decision Letter · Decision Letter 1]

31 Dec 2019

PONE-D-19-26580R1

Human and climatic drivers affect spatial fishing patterns in a multiple-use marine protected area: the Galapagos Marine Reserve

PLOS ONE

Dear Dr. Castrejón,

Thank you for submitting your manuscript to PLOS ONE. After careful consideration, we feel that it has merit but does not fully meet PLOS ONE’s publication criteria as it currently stands. Therefore, we invite you to submit a revised version of the manuscript that addresses the points raised during the review process.

The manuscript is much improved so I thank the authors for their efforts and Reviewer 2 is satisfied that all the suggestions have been address (Reviewer 1 was not available). That said, Reviewer 2 has noted that the Discussion still requires some work and I agree. The authors should emphasis the important findings first, rather than leading with the 'fishing the line' story. I still think the Discussion could be trimmed down to be more concise as well so I encourage the authors to make their revisions with that in mind. I have some minor editorial corrections as well.

That said, I think these changes can be made relatively quickly and that a more focused and concise Discussion will improve the paper. Once these changes have been made the paper can be accepted. I look forward to seeing the final version of this interesting and timely paper in the near future.

We would appreciate receiving your revised manuscript by Feb 14 2020 11:59PM. To enhance the reproducibility of your results, we recommend that if applicable you deposit your laboratory protocols in protocols.io, where a protocol can be assigned its own identifier (DOI) such that it can be cited independently in the future. For instructions see: http://journals.plos.org/plosone/s/submission-guidelines#loc-laboratory-protocols

We look forward to receiving your revised manuscript.

Kind regards,

Heather M. Patterson, Ph.D.

Academic Editor

PLOS ONE

Reviewers' comments:

Reviewer's Responses to Questions

**Comments to the Author**

1. If the authors have adequately addressed your comments raised in a previous round of review and you feel that this manuscript is now acceptable for publication, you may indicate that here to bypass the “Comments to the Author” section, enter your conflict of interest statement in the “Confidential to Editor” section, and submit your "Accept" recommendation.

Reviewer #2: All comments have been addressed

2. Is the manuscript technically sound, and do the data support the conclusions?

Reviewer #2: Partly

3. Has the statistical analysis been performed appropriately and rigorously? 

Reviewer #2: Yes

4. Have the authors made all data underlying the findings in their manuscript fully available?

Reviewer #2: Yes

5. Is the manuscript presented in an intelligible fashion and written in standard English?

Reviewer #2: Yes

6. Review Comments to the Author

Reviewer #2: I commend the authors for the considerable effort they have put into their revisions. Points where they opted not to make changes were adequately explained.

However, the Discussion is still a bit of a mess. Somehow now the discussion starts with the "fishing the line" story, which is not really one of the main points this paper is making. The following paragraphs then jump around various topics, and the main points are mostly still lost in the detail. My suggestion is: 1) Present all the broad topics and key points in the opening paragraph. 2) Before writing the text, list all the key points and sub-points in order of importance (or sequentially as they appear in the results section. 3) Organize the text accordingly. The content is all there, it just needs to be organized and edited for flow. It's also still very long. Consider what sentences you could lose without affecting the content or the message.

7. PLOS authors have the option to publish the peer review history of their article (what does this mean?). If published, this will include your full peer review and any attached files.

Reviewer #2: No

---

## [Author Response · Author response to Decision Letter 1]

6 Jan 2020

January 5th, 2020

Heather M. Patterson, Ph.D.

Academic Editor 

PlosOne

Dear Dr. Patterson,

Please find enclosed the revised version of the manuscript entitled “Human and climatic drivers affect spatial fishing patterns in a multiple-use marine protected area: the Galapagos Marine Reserve” by Mauricio Castrejón (corresponding author) and Anthony Charles. 

As required, a revised version of our manuscript that addresses the points raised during the review is submitted as a single Word document file. Given the novel nature of the topic and lack of previous long-term quantitative studies on Galapagos marine zoning, the paper required a longer treatment that the normal article length for PlosOne. 

Also please find attached a rebuttal letter that contains our responses to each point raised by the academic editor and reviewers. We are grateful for the insightful and constructive comments made by all of you. We accepted all your suggestions. We have written the discussion in a more focused and concise manner, as requested.

The Galapagos National Park Service is the owner of the data used for this paper. Therefore, there are legal restrictions on sharing a de-identified data set. However, the data underlying the results presented in the study are available on request at investigacion@galapagos.gob.ec. We confirm that other researchers would be able to access the data set in the same manner as we did, and we did not have any special access privileges that others would not have.

We hope you will be pleased with this new version of the manuscript. Please do not hesitate to contact me with any questions or concerns. Thank you very much for your consideration. We look forward to receiving the acknowledgment of the manuscript and your decision.

Sincerely yours,

Dr. Mauricio Castrejón

Interdisciplinary PhD Program

Dalhousie University

Halifax, Canada

mauricio.castrejon@dal.ca

PONE-D-19-26580R1

Human and climatic drivers affect spatial fishing patterns in a multiple-use marine protected area: the Galapagos Marine Reserve 

Rebuttal letter

We acknowledge useful comments provided by the academic editor and one referee. All their suggestions have been included in the revised version of the manuscript. In this letter, we explain how and where the corrections have been incorporated into the text. We follow the editorial order and have numbered referees' suggestions and comments.

Editorial comments

The manuscript is much improved, so I thank the authors for their efforts and Reviewer 2 is satisfied that all the suggestions have been address (Reviewer 1 was not available). That said, Reviewer 2 has noted that the Discussion still requires some work and I agree. The authors should emphasis the important findings first, rather than leading with the 'fishing the line' story. I still think the Discussion could be trimmed down to be more concise as well so I encourage the authors to make their revisions with that in mind. I have some minor editorial corrections as well.

That said, I think these changes can be made relatively quickly and that a more focused and concise Discussion will improve the paper. Once these changes have been made the paper can be accepted. I look forward to seeing the final version of this interesting and timely paper in the near future. 

R: We have written the Discussion section in a more focused and concise manner, emphasizing the most important findings first. We shifted the material around and consolidated the discussion into four main subsections. The first of these is about the 'big picture' of change in the Galapagos, in response to the drivers of change analyzed. The second subsection focuses on the detailed assessment of what caused dynamics of fishing effort. The third brings together all the no-take zones analysis. The fourth summarizes the conclusions and lessons learned. The whole Discussion section was edited and reduced. 

Minor Comments

Line 30: Rewrite as ‘Assessments of the effectiveness of marine protected areas (MPAs)’

 R: Suggestion taken. 

Line 65: Write as ‘particularly those designed for multiple use’ (no hyphen in multiple use here’

R: Suggestion taken.

Line 67: I think this should be ‘in which as assessment of the performance of MPAs’

R: Suggestion taken.

Line77: Should be ‘processes’

R: Suggestion taken.

Line 146: Do not introduce the abbreviation ‘ENSO’ as it is only used once again in the manuscript. It is PLoS formatting that abbreviations must be used at least 3 times if they are introduced.

R: Suggestion taken.

Line 185: Just abbreviate the scientific name here as P. penicillatus as the full name has already been provided

R: Suggestion taken.

Line 194: I would use MPA rather than marine reserve here to be consistent in the terminology.

R: Suggestion taken.

Line 234: Do not introduce the abbreviation ‘GPS’ as it is not used again in the manuscript.

R: Suggestion taken.

Line 309: What is ‘GNP’? This has not been defined yet. I would just spell it out.

R: Suggestion taken.

Line 403: Do not introduce the abbreviation ‘ENSO, just spell it out.

R: Suggestion taken.

Line 1057: Should be ‘shellfish fisheries’

R: Suggestion taken.

---

## [Editor Report · Decision Letter 2]

8 Jan 2020

Human and climatic drivers affect spatial fishing patterns in a multiple-use marine protected area: the Galapagos Marine Reserve

PONE-D-19-26580R2

Dear Dr. Castrejón,

We are pleased to inform you that your manuscript has been judged scientifically suitable for publication and will be formally accepted for publication once it complies with all outstanding technical requirements.

With kind regards,

Heather M. Patterson, Ph.D.

Academic Editor

PLOS ONE
---

## [Editor Report · Acceptance letter]

14 Jan 2020

PONE-D-19-26580R2 

Human and climatic drivers affect spatial fishing patterns in a multiple-use marine protected area: the Galapagos Marine Reserve 

Dear Dr. Castrejón:

I am pleased to inform you that your manuscript has been deemed suitable for publication in PLOS ONE. Congratulations! Your manuscript is now with our production department. 

With kind regards,

on behalf of

Dr. Heather M. Patterson 

Academic Editor

PLOS ONE